# Medial geniculate body and primary auditory cortex differentially contribute to striatal sound representations

Liang Chen[1], Xinxing Wang[1], Shaoyu Ge[1] & Qiaojie Xiong [1]

The dorsal striatum has emerged as a key region in sensory-guided, reward-driven decision making. A posterior sub-region of the dorsal striatum, the auditory striatum, receives convergent projections from both auditory thalamus and auditory cortex. How these pathways contribute to auditory striatal activity and function remains largely unknown. Here we show that chemogenetic inhibition of the projections from either the medial geniculate body (MGB) or primary auditory cortex (ACx) to auditory striatum in mice impairs performance in an auditory frequency discrimination task. While recording striatal sound responses, we find that transiently silencing the MGB projection reduced sound responses across a wide-range of frequencies in striatal medium spiny neurons. In contrast, transiently silencing the primary ACx projection diminish sound responses preferentially at the best frequencies in striatal medium spiny neurons. Together, our findings reveal that the MGB projection mainly functions as a gain controller, whereas the primary ACx projection provides tuning information for striatal sound representations.

[1] Department of Neurobiology & Behavior, Stony Brook University, Stony Brook, NY 11794, USA. Correspondence and requests for materials should be addressed to Q.X. (email: qiaojie.xiong@stonybrook.edu)

The neostriatum, the main input structure of the basal ganglia, has been widely implicated in habitual and social behaviors[1–3], decision-making[4–6], and reinforcement learning[7]. Striatal neurons can respond to visual, auditory, or somatosensory stimulation[8], consistent with anatomically demonstrated sensory inputs. The dorsal striatum receives topographically organized projections from nearly the entire neocortex[9], and from most thalamic nuclei[10,11]. The axon terminals of cortical and thalamic projections converge with comparable densities onto individual striatal neurons, forming functional glutamatergic synapses (i.e., thalamostriatal and corticostriatal synapses)[12–14]. How these two projections coordinate to regulate striatal activities and striatal-dependent behaviors remain largely unknown.

Previous studies suggest that thalamostriatal projections may be important for alertness and behavioral switching[15]. However, the mechanisms underlying this function remain unclear. Moreover, the majority of previous studies have focused on the motor subdivision of the dorsal striatum[16–19]. The sensory subdivision, especially the auditory striatal region which receives projections from the medial geniculate body (MGB, the main auditory thalamus) and the auditory cortex (ACx), remains largely elusive.

Two recent studies have found that projections from ACx to the auditory striatum drives decision-making in rodents, and that selective plasticity of these synapses may underlie the establishment of this behavioral association[20,21]. To understand how MGB projection to the auditory striatum contribute to auditory decision-making, we used chemogenetic tools to assess the effects of the MGB projection inhibition on performance of an auditory frequency-discrimination task. Upon demonstrating the behavioral relevance of this connection, we used in vivo tetrode recordings to characterize striatal sound responses to the stimuli used in the behavioral task, and then optogenetically dissected the MGB and primary ACx contributions to the striatal sound representation. Our results indicate that the projections from the MGB and primary ACx differentially modulate striatal auditory information and that these effects are essential for making an auditory frequency-discrimination decision.

## Results

### Suppression of the MGB projection impairs task performance.

The auditory striatum receives convergent projections from the MGB and primary ACx (Fig. 1a and Supplementary Fig. 1). The corticostriatal projection has recently been shown to drive auditory decisions in an auditory frequency-discrimination "cloud-of-tones" task[20]. In this study, we examined whether and how MGB and primary ACx projections modulate auditory decision-making in "cloud-of-tones" task.

To assess the effects of the projections on the "cloud-of-tones" task, we used a chemogenetic method to selectively and reversibly inhibit the MGB and primary ACx projections individually. We first expressed hM4Di-mCherry, an inhibitory DREADD (*Designer Receptors Exclusively Activated by Designer Drug*[22]), in MGB neurons via adeno-associated virus (AAV) infection (Fig. 1b, left). Five weeks after viral infection, we observed strong mCherry signals of the MGB projection in the auditory striatum (Fig. 1b, right panel). We then tested the inhibition effect of this projection mediated by the hM4Di receptor. In a validation experiment, we co-injected AAV expressing Channelrhodopsin 2 (ChR2) with AAV expressing hM4Di into the MGB, and delivered blue light pulses to activate ChR2-positive axon terminals on acutely prepared striatal slices. We applied the synthesized activator, clozapine-N-oxide (CNO), into the bath solution after establishing whole-cell patch recordings on striatal

neurons. We found that the addition of CNO (10 μM) substantially suppressed the light-evoked postsynaptic responses as shown in Supplementary Fig. 2A. To test any potential side effects due to nonspecific binding of CNO to other endogenous receptors[23], we conducted the same experiment on control slices prepared from mice expressing only mCherry, but not hM4Di. We observed no detectable effects on synaptic transmission at the CNO concentration used here (10 μM) (Supplementary Fig. 2B). The ex vivo efficacy of hM4Di-mediated inhibition of the MGB projection to the auditory striatum suggested that a similar strategy may work in vivo.

We then assessed whether inhibition of the MGB projection affects auditory decision-making in the "cloud-of-tones" task. We bilaterally expressed hM4Di-mCherry or mCherry alone (as control) in MGB neurons in mice via AAV infection. Two weeks after the viral infection, we trained these mice to perform the "cloud-of-tones" task. In brief, a freely moving mouse was placed in a sound-proof chamber (as previously described[21]). Each trial is self-initiated by the mouse poking its nose in the center port to trigger a sound stimulus, and the mouse learns to associate the frequency of pure tones (high versus low) with actions (going to the left versus the right port) for a water reward through trial-and-error (Fig. 1c). We then bilaterally implanted cannulas into the auditory striatum for local CNO or saline infusion (Fig. 1b and Supplementary Fig. 3). We continued to train the mice on task after they recovered from the implantation surgery (around 2 weeks), and initiated chemogenetic manipulation to inhibit the MGB projection while the mice performed the task. We then compared individuals' performances during saline infusion sessions (the sessions immediately before the CNO sessions) to their performance during CNO infusion sessions in both hM4Di-mCherry and control (mCherry only) groups. In hM4Di-mCherry mice, we observed that the CNO infusion substantially decreased choice accuracy in task performance as compared to saline infusion sessions (Fig. 1d, e). In contrast, we found no observable effect upon CNO application in control (mCherry only) mice (Fig. 1d, e).

To further clarify how inhibiting MGB striatal projections negatively influenced task performance, we performed the following analyses: we first quantified trials in which the sound stimuli were triggered but the mice did not choose either of the side ports. We did not observe differences in the numbers of these nonreport trials across control or CNO sessions (Fig. 1f). Next, we counted the number of completed trials per session and observed no differences among control or CNO sessions (Fig. 1g). Furthermore, there were no change in reaction times (response latency) across difficulties (Fig. 1h). These analyses confirmed that the CNO application did not affect the motivational state or motor capabilities of our experimental mice.

We applied the same strategy to examine the behavioral impacts of the corticostriatal pathway from the primary ACx to the auditory striatum. Our analysis showed that inhibition of this cortical projection also significantly decreased choice accuracy in task performance, but had no effects on reaction time (Supplementary Fig. 4).

Together, our results revealed that hM4Di-mediated inhibition of the MGB or the primary ACx projection reduced choice accuracy across different difficulties, indicating that both thalamic and cortical projections to the auditory striatum are involved in decision-making in the "cloud-of-tones" task.

### Characterization of striatal responses to the tone stimuli.

The findings that both MGB and primary ACx projections to the auditory striatum are critical for auditory decision-making led us to ask what type of auditory information these two projections

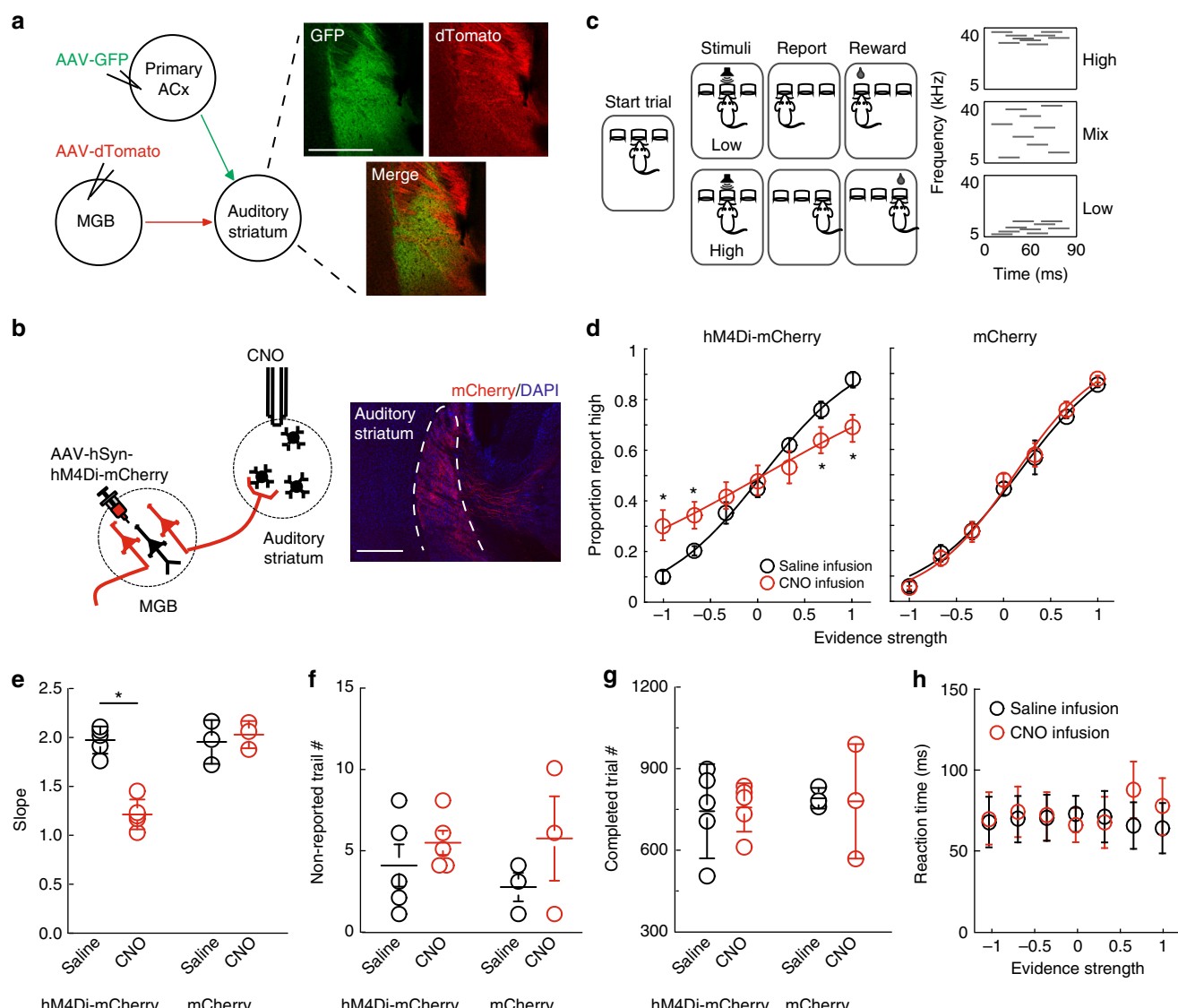

**Fig. 1** Inhibition of the MGB projection to the auditory striatum impaired animals' performance in "cloud-of-tones" task. **a** Left panel, illustration of the projections from the MGB and primary ACx to the auditory striatum. Right panel, Example images of thalamic projections (green) and cortical projection (red) to the auditory striatum. AAV expressing GFP was injected into MGB, and AAV-expressing dTomato was injected into auditory cortex. Images are taken at the auditory striatum. Scale bar, 500 μm. **b** CNO/DREADD-mediated inhibition of thalamostriatal projection. Left panel, schematic diagram of viral injection and CNO infusion setup. Right upper panel, example image of labeled thalamostriatal fibers. Scale bar: 100 μm. Right lower panel, one example of whole-cell recordings (average EPSCs) on striatal slice showing CNO-mediated terminal inhibition. EPSCs on MSN was elicited by local electrical stimulation on axon fibers, with CNO (10 μM, red) and without CNO (black) application. Bicuculine (20 μM) was added in the bath solution to block local GABAergic activity. Scale bars: 5 pA and 20 ms. **c**. Left panel, illustration of "cloud-of-tones" task, a two-alternative forced-choice frequency-discrimination task. Right panel, examples of auditory stimuli. **d** CNO effects on task performance in experimental mice (expressing hM4Di-mCherry, left) and control mice (expressing mCherry only, right). The evidence strength is calculated as (# high tones − # low tones) / (# high tones + # low tones). Error bars are s.e.m. The curve is fitted with logistic sigmoid function. Red: CNO sessions; Black: saline sessions. **e–h**. Effects of CNO-mediated thalamic inhibition on the slopes of psychometric function (**e**), the numbers of non-reported trials per session (**f**), the number of completed trials per session (**g**), and the reaction time (**h**). For **d–h**, $n = 5$ pairs of sessions from three experimental mice, and three pairs of session from three control mice; error bars are s.e.m., $^*p < 0.05$, paired $t$ test

carry. To answer this question, we first characterized the striatal responses to sound stimuli. In the "cloud-of-tones" task, animals choose side ports based on judgements about the frequencies of the pure tones in the stimuli. We therefore used pure tones for this test. We presented 100 ms pure tones at different frequencies (2–50 kHz) to the freely moving animals inside a sound-proof chamber. We implanted tetrodes into the left auditory striatum to record tone-evoked responses from striatal neurons. There are three typical types of neurons in the dorsal striatum: medium spiny neurons (MSNs) make up about 95% of the neuronal population, and the remainder includes cholinergic interneurons (ChI) and fast-spiking interneurons (FS)[24]. These three types can be readily distinguished by their spike waveforms from tetrode recordings (Fig. 2a, detail in Methods) as previously described[17,25]. In our tetrode recordings, 80.1% (944/1178) of neurons were identified as MSNs, 12.9 % (152/1178) were identified as FSs, and 7% (82/1178) were identified as ChIs. This closely resembled the proportions of recordings reported in previous studies[26,27]. We further analyzed the tone-evoked responses in each type of striatal neuron, and found 206 of

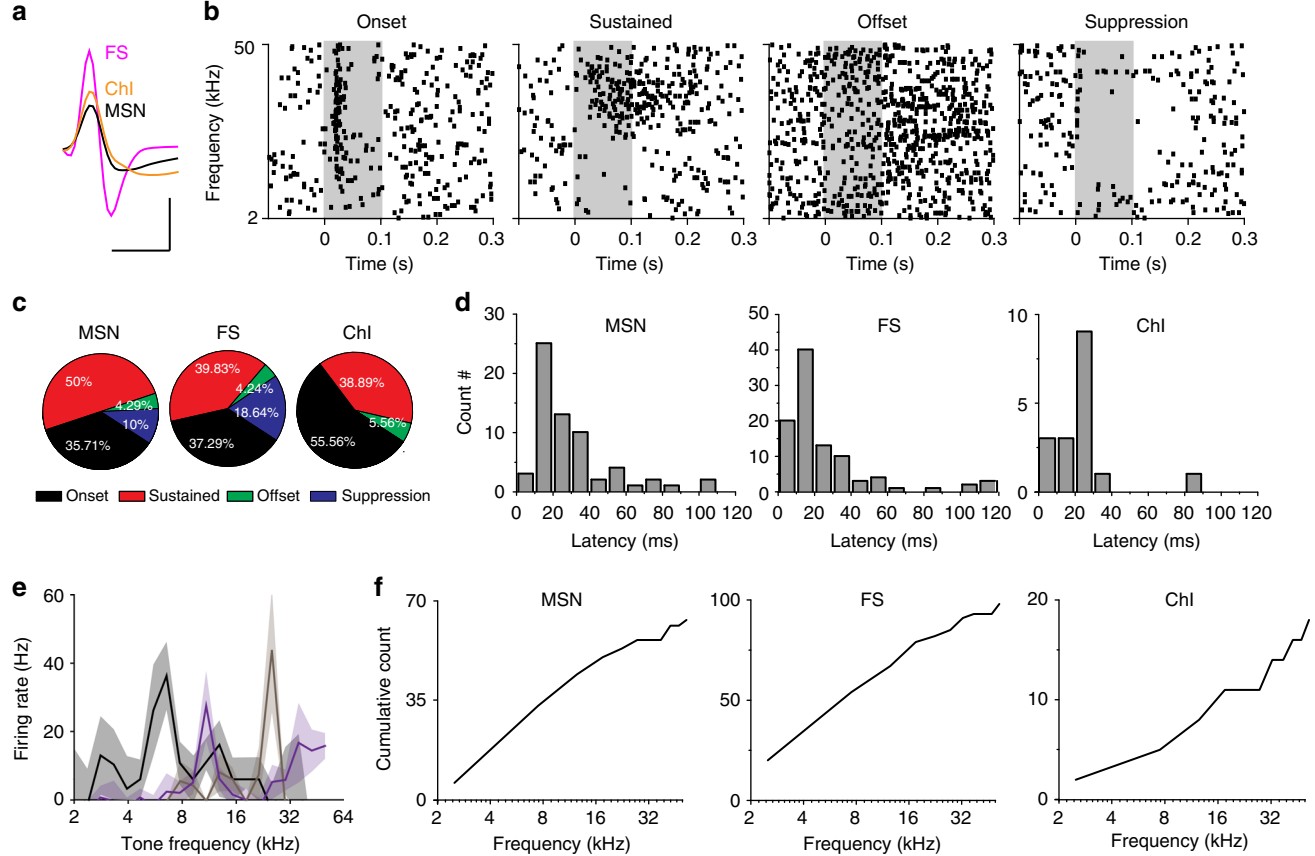

**Fig. 2** Striatal neurons respond to tonal stimuli. **a** Example of isolated striatal single units for different cell types based on half-valley width of waveforms. Scale bars: 0.5 ms and 50 μV. **b** Example of striatal sound responses in raster plots. Individual pure tones ranged from 2 to 50 kHz are presented at 0–0.1 s time window (shaded area). **c** Portion of each type of tonal response for individual striatal cell types. **d** Distribution of response latency for individual striatal cell types. Latency was calculated as the time point when firing rate is significantly higher than the baseline (average firing rate of 0–50 ms before the onset of tones). **e** Examples of three tuning plots. Averaged tone-evoked firing rates are plotted against the corresponding tone frequencies. Solid lines: mean values; shaded areas: standard errors. **f** Accumulative curves of best frequencies for individual striatal cell types. For **c–f**: MSN, $n = 70$; FS, $n = 118$; ChI, $n = 18$; from 14 mice

1178 recorded neurons responded to tonal stimuli (MSN:70/944; FSI:118/152; ChI:18/82) (see Methods for tone responsiveness criteria).

All three types of striatal neurons displayed diverse responses to pure tones, with some neurons exhibiting rapid onset of transient responses while others showed sustained responses, offset responses or suppression responses (Fig. 2b, detail in Methods). The majority of cells in all three cell types manifested transient onset responses, followed by sustained and suppression responses (Fig. 2c). MSNs and FS had a wide range of response latencies from 10 to 100 ms, while ChI had latencies around 30 ms (Fig. 2d).

Striatal neurons responded to tone stimuli over a wide range of frequencies (Fig. 2b). Among the tone-responsive neurons, 160/ 206 neurons were tuned (see Methods for criteria). We only included tuned neurons in later tuning analysis. Note that in our recordings, most striatal neurons did not respond to tonal stimuli lower than 50 dB, and there was also a large portion of neurons that did not respond to 60 dB. We thus used either 60 or 70 dB (the lower responsive intensity, see Methods for details) stimuli to calculate the best frequency for each striatal neuron (the frequency at which the neuron has its highest peak firing rate) (Fig. 2e). We found that neurons with onset and sustained responses displayed best frequencies to the pure tones over the full range of tested stimuli (Fig. 2f), while most of the units with suppression responses were more broadly tuned (Supplementary

Fig. 9B left panel). Together, our profiling revealed that all three major types of striatal neurons responded to tonal stimuli, and showed frequency preferences.

We next asked from where the auditory striatum receives the sound information. As shown in the Fig. 1a, the auditory striatum receives projections from both the primary ACx and MGB. Since the primary ACx also receives ascending projections from the MGB[28–30], we asked whether the MGB neurons that project to the ACx and striatum are from the same or segregated populations (Supplementary Fig. 5A). To determine the projection patterns of MGB neurons, we retrogradely labeled the MGB neurons that project to the primary ACx and the auditory striatum with a retrograde reagent (cholera toxin B, CTB). Specifically, as shown in Supplementary Fig. 5B, we injected CTB conjugated with a red or green fluorophore into the primary ACx and the auditory striatum, respectively. Seven days after the injection, we collected brain tissue and analyzed the images. We found that the dorsal MGB (MGd) preferentially projected to the auditory striatum, while those neurons projecting to the primary ACx are mainly in the ventral MGB (MGv) (Supplementary Fig. 5B–D). These anatomical tracing studies showed that the auditory striatum receive MGB and the primary ACx projections with different origins in the MGB. Interestingly, several previous studies have reported that neurons in the MGd are broadly tuned to tonal stimuli, while MGv neurons are sharply tuned[30–33]. Moreover, the tonotopic organization is preserved in the MGv[34],

the primary ACx[30,35,36], and the primary ACx projections to the striatum[21]. These studies, together with our anatomical tracing, suggest that the two types of projections may likely relay distinct acoustic information to the auditory striatum in regulating striatal-dependent auditory decisions.

**Silencing MGB projection suppressed striatal sound responses.** To test this hypothesis, we selectively and transiently silenced the MGB projection and recorded the striatal sound responses. We expressed the neural silencer archaerhodopsin (ArchT)[37] in MGB neurons via AAV injection. We then implanted a microdrive with integrated optic fibers and tetrodes into the auditory striatum for sound response recording and thalamic projection silencing (Fig. 3a). Similar to the experimental design in Fig. 2, we presented pure tones to the mice, and recorded the sound responses of striatal neurons. In half of the randomly selected trials, we

delivered light pulses through the optic fiber that began 100 ms prior to the onset of tones and ended 100 ms after the offset of tones, to activate ArchT during tone presentation (validation in Supplementary Fig. 6, details in Methods).

After sorting the recordings based on cell type, we focused on analyzing the responses of MSNs in this study, as they are the most common neuron type, and the only projection neurons in the dorsal striatum[24]. We compared striatal single-unit responses to sound stimuli between trials with and without light stimulation (i.e., with and without inhibition of thalamic projections). Our "cloud-of-tones" task is a frequency-discrimination task, thus we focused our analysis on two features of the sound responses: amplitude and frequency tuning. We found that silencing the MGB projection substantially decreased the overall amplitude of tone-evoked responses of MSNs (Fig. 3b and Supplementary Fig. 8A). Silencing thalamic projections, however, did not alter

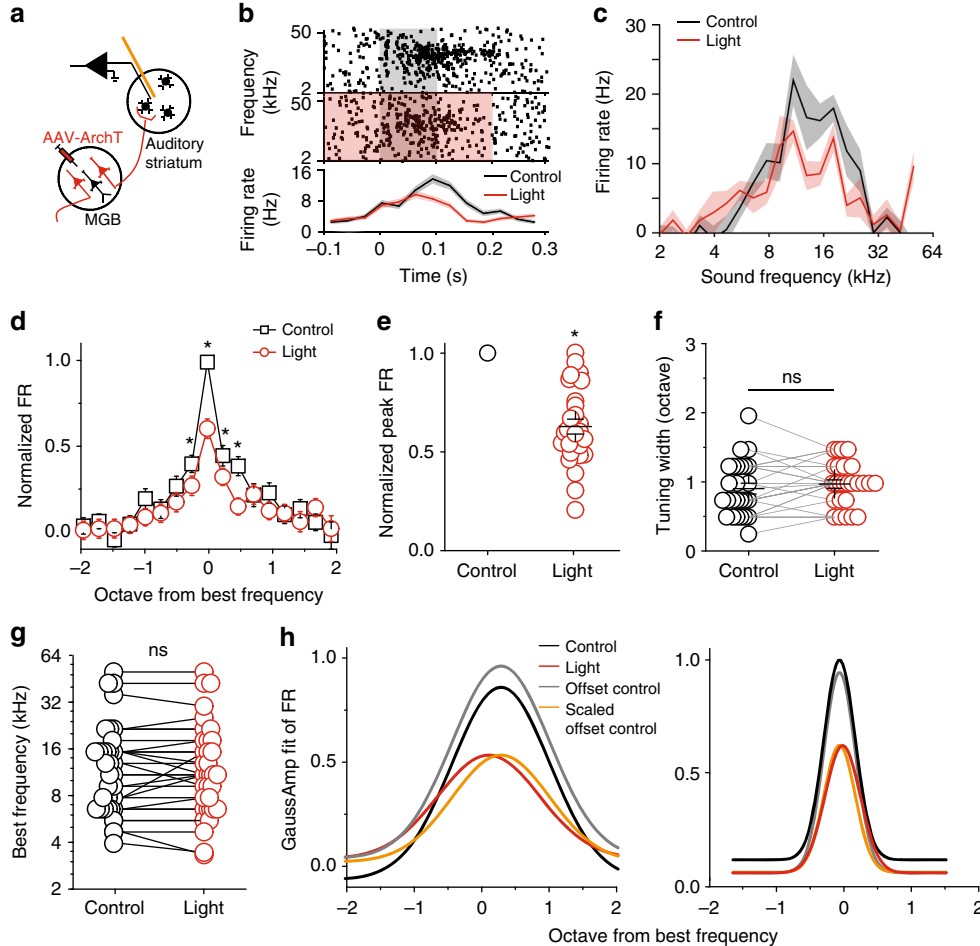

**Fig. 3** Silencing the MGB projection to the auditory striatum reduced striatal MSN tonal responses across frequencies. **a** Illustration of recording setup for silencing MGB projection. **b** One example of recorded MSN single unit. Upper panels, raster plots of a MSN firings during tone presentation (gray shaded area) and tone + light presentation (red + gray shaded area). Tone: 0–0.1 s time window; Light: −0.1 to 0.2 s time window. Lower panel, peristimulus time histogram (PSTH) of the firing rates in response to tone alone (black) and tone + light (red). **c** Tuning plots of the example shown in (**b**). **d** Average tuning plots at the tone alone (black) and tone + light (red) conditions. Frequencies are aligned to the best frequency of each single unit and scaled in octave away from the best frequency. The best frequencies are determined separately from control and light-on trials for each single unit. Firing rates at different frequencies of each single unit are normalized to the firing rate at the best frequency from control trials. **e** Normalized individual and averaged firing rates at best frequencies at the tone alone (black) and tone + light (red) conditions. **f** The tuning widths (the frequency range at half peak firing rate) at the tone alone (black) and tone + light (red) conditions. **g** The best frequencies at the tone alone (black) and tone + light (red) conditions. **h** The fitted tuning curves with a Gaussian function ($R(f) = A \times \exp(-0.5 \times (\theta - f_0)^2/\sigma^2) + B$, where B represents the baseline response, A represents the amplitude of the strongest evoked response, $f_0$ represents the preferred frequency, and $\sigma$ is the tuning width). Left panel: the fitted curves from the same example neurons shown in (**b** and **c**). Right panel: the fitted curves from the population data shown in (**d**). Black, control condition; red, light-on condition; gray, offset the black curve to the baseline of the red curve; orange, scaled down the gray curve. For **d**–**h**, n = 26 neurons from five mice, data are presented as mean values, error bars are s.e.m., * p < 0.01, t test in (**d**) and paired t test in (**e**–**g**)

the best frequency of individual striatal single units, or the tuning width (Fig. 3c–g). We further analyzed the tone-evoked responses by fitting the tuning curves with a Gaussian function, and found that the fitted tuning curves in control condition could be transformed to those in light-on condition by subtracting a small offset (0.01 ± 0.02) and dividing a scale factor (1.8 ± 0.2) as shown in Fig. 3h. These findings suggested that MGB projection to the auditory striatum provided a gain controlling function to the tone-evoked responses.

**Silencing the ACx projection reduced striatal sound tuning.** To determine the sound information that is relayed by the primary ACx projections to the auditory striatum, we performed the same tests shown in Fig. 3, but with silencing of the corticostriatal pathway instead. In brief, we expressed ArchT in primary ACx neurons via AAV injection, and implanted a microdrive with integrated optic fibers and tetrodes into the auditory striatum for recording and cortical projection silencing (Fig. 4a). We performed similar analyses as described above for the MGB pathway studies. We found that silencing the primary auditory cortical projection also substantially decreased the overall amplitude of tone-evoked responses of MSNs (Fig. 4b). Interestingly, unlike silencing of the MGB projection—which led to a decrease to broad frequencies in responsiveness (Fig. 3d)—silencing of the primary auditory cortical projection preferentially decreased the responses of individual striatal units to their best frequencies (Fig. 4c–e and Supplementary Fig. 8B), but not the shoulder frequencies (Fig. 4f). Together, these results suggest that the primary ACx projection provides tone-tuned information to the auditory striatum.

**Both pathways comprise the major acoustic sources.** The finding that MGB and primary ACx projections carry differential sound information motivated us to extend our analysis to test whether it would be sufficient to block sound response by silencing both projections. As shown in Supplementary Figure 1, the MGB and the primary ACx are the two primary auditory areas projecting to the auditory striatum. To test this possibility, we expressed the neuronal silencer ArchT in both MGB and the primary ACx, followed by implanting tetrode bundles with an optic fiber into the auditory striatum to record tone-evoked responses. We performed similar tests as described in Figs. 3 and 4. We found that optical silencing of both MGB and primary ACx terminals largely abolished the striatal sound responses (Fig. 5a–d). This indicates that the MGB and the primary ACx are the two major sources of acoustic information to the auditory striatum. Together with those results from the selective inhibition of either MGB (Fig. 3) or the primary ACx (Fig. 4) projections, we propose that the primary ACx projection relay tuning information of the tonal stimuli to striatum, while the MGB projection functions as a gain controller across a broad array of frequencies (Fig. 5e).

## Discussion

In this study, using an established auditory frequency-discrimination task, we found that chemogenetic inhibition of the MGB or the primary ACx projection to the auditory striatum impaired task performance. In an effort to examine the underlying mechanism, we found that optogenetic silencing of the MGB projection alone reduced the amplitudes of tone-evoked responses in MSNs across different frequencies. In contrast,

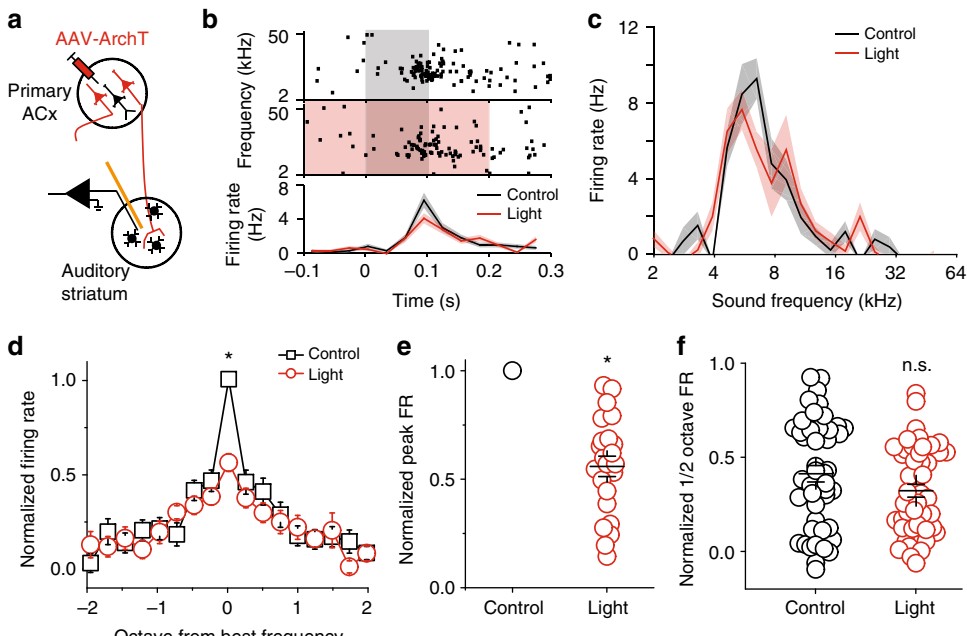

**Fig. 4** Silencing the primary ACx projection to the auditory striatum reduced striatal MSN tonal responses to their best frequencies. **a** Illustration of recording setup for silencing primary ACx projection. **b** One example of recorded MSN single unit. Upper panels, raster plots of a MSN firings during tone presentation (gray shaded area) and tone + light presentation (red + gray shaded area). Tone: 0–0.1 s time window; Light: −0.1–0.2 s time window. Lower panel, PSTH of the firing rates in response to tone alone (black) and tone + light (red). **c** Tuning plots of the example shown in (**b**). **d** Average tuning plots at the tone alone (black) and tone + light (red) conditions. Average tuning plots at the tone alone (black) and tone + light (red) conditions. Frequencies are aligned to the best frequency of each single unit and scaled in octave away from the best frequency. The best frequencies are determined separately from control and light-on trials for each single unit. Firing rates at different frequencies of each single unit are normalized to the firing rate at the best frequency from control trials. (**e**) Normalized individual and averaged firing rates at best frequencies at the tone alone (black) and tone + light (red) conditions. (**f**) Normalized individual and averaged firing rates at frequencies ± ½ octave away from the best frequency at the tone alone (black) and tone + light (red) conditions. For **d**–**f**, $n = 22$ neurons from five mice, data are presented as mean values, error bars are s.e.m., * $p < 0.01$, $t$ test in (**d**) and paired $t$ test in (**e** and **f**)

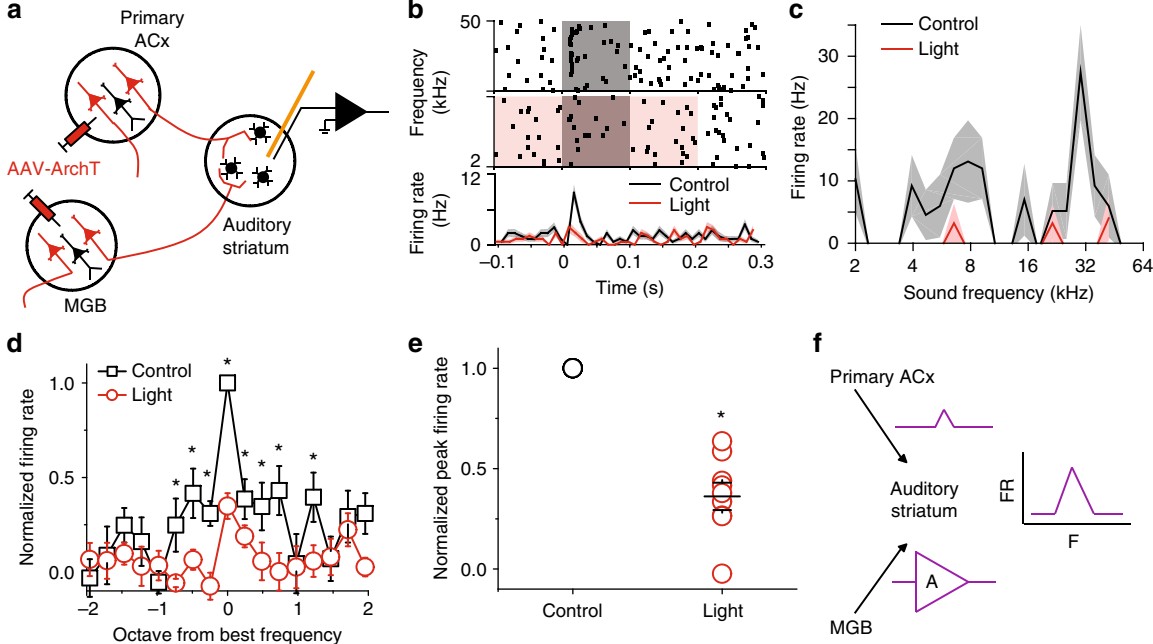

**Fig. 5** MGB and primary ACx projections are the two major sources for striatal tonal response. **a** Illustration of recording setup for silencing both MGB and primary ACx projections. **b** One example of recorded MSN single unit. Upper panels, raster plots of a MSN firings during tone presentation (gray shaded area) and tone + light presentation (red + gray shaded area). Tone: 0–0.1 s time window; Light: −0.1 to 0.2 s time window. Lower panel, PSTH of the firing rates in response to tone alone (black) and tone + light (red). **c** Tuning plots of the example shown in (**b**). **d** Average tuning plots at the tone alone (black) and tone + light (red) conditions. Frequencies are aligned to the best frequency of each single unit and scaled in octave away from the best frequency. The best frequencies are determined separately from control and light-on trials for each single unit. Firing rates at different frequencies of each single unit are normalized to the firing rate at the best frequency from control trials. **e** Normalized individual and averaged firing rates at best frequencies at the tone alone (black) and tone + light (red) conditions. For **d**, **e**, $n = 9$ neurons from two mice, data are presented as mean values, error bars are s.e.m., * $p < 0.01$, t test. **f** Proposed model of MGB and primary ACx contributions to striatal tonal responses

silencing the primary ACx projection preferentially reduced tone-evoked responses of the MSNs' at their best frequencies. In sum, our results reveal that the MGB and the primary ACx projections to the auditory striatum relay differential sound information, that is necessary for decision-making in the "cloud-of-tones" task.

In sensory perception, it has been proposed that there are two parallel pathways: the driver (first order) and the modulator (higher order)[38,39]. For auditory processing, the driver pathway is thought to extend from the peripheral receptors, to the central nucleus of inferior colliculus, to the ventral MGB and then to the primary ACx. The modulator pathway refers to inputs that the dorsal and medial MGB receives from both the inferior colliculus and the primary ACx, and the MGB's projections to the secondary ACx and limbic related areas[40]. The driver pathway carries primary, high-fidelity information, and defines the essential activity patterns; while the modulator pathway controls the transmission gain of information without contributing significantly to the general pattern[41]. In the present study, we investigated two parallel auditory inputs to the auditory striatum, one from the primary ACx that receives projections from the ventral MGB (in the driver pathway), and one from the dorsal MGB (in the modulator pathway). We found that inhibition of these two auditory inputs impaired auditory decision-making in a tone-discrimination "cloud-of-tones" task (Fig. 1 and Supplementary Fig. 4). Furthermore, silencing the MGB projection to the auditory striatum dampened tone-evoked responses across broad tonal frequencies in auditory striatum, whereas silencing the ACx projection preferentially diminished the tuning (Figs. 3 and 4). Our behavioral and physiological discoveries thus support the "driver/modulator" model, and these findings further extend the understanding of these pathways in auditory processing.

We should point out that the auditory striatum also receives direct projections from other auditory thalamic nuclei, such as peripeduncular nucleus (ref. 30, also shown in Supplementary Fig. 5B), and secondary ACx (Supplementary Fig. 1B). These projections could be part of the modulatory pathways. The functions of these projections, and how they fit in the "driver/modulator" model require further investigation.

Both the primary ACx neurons and their projections to the auditory striatum are tonotopically organized[21,30,36]. If inhibiting only part of the primary ACx leads to the preferential suppression of the peak response, we would expect that the striatal neurons shown suppression effects shared the same best frequencies. However, within one animal we identified tuning effects from MSNs with very different best frequencies, suggesting that the preferential suppression of peak response is likely not due to insufficient cortical infection coverage.

We found that MGB and primary ACx projections relayed different sound information to the auditory striatum. However, it remains unclear where the sound information from these two projections is integrated: does this occur through striatal network activity or within individual striatal MSNs? In our current study, we focused on analyzing the MSN neurons. It remains to be tested if and how FS and ChI are involved in the local circuit to modulate MSN responses to thalamic and cortical inputs? In vivo or in vitro whole-cell recordings of auditory striatal neurons will be needed to understand the cellular and subcellular mechanisms underlying the striatal integration of thalamic and cortical inputs. Furthermore, among the tetrode recording identified neurons, 7.4% (70/944) MSNs, 77.6% (118/152) FSs, and 22% (18/82) ChIs showed tone responses. There was a smaller percentage of MSNs responding to tonal stimulation. This may be due to unique

circuitry or cellular mechanisms, which is also an interesting question to explore in future study.

In the current study, our analyses focused on the MSNs with onset and sustained responses (Supplementary. Fig. 8). Due to the limited samples (Supplementary. Fig. 9), it remains unclear how MGB and ACx projections modulate the offset and suppression responses of MSNs. These types of MSNs may or may not receive direct projections from the MGB and the primary ACx. How they respond to tones and how they are involved in striatal local circuits need further examinations.

MGB and primary ACx inputs differentially modulate striatal sound responses, which are important for decision-making in the "cloud-of-tones" task. In this study, we focused on determining how these two projections carry sound information to the auditory striatum. It remains unclear how striatal sound responses are linked to motor planning and initiation for behavioral choices in this task. Together with a previous study[21], we proposed a simplified model here (Supplementary Fig. 10). In well-trained animals that learned to associate low-frequency tones with rightward choice (for example), the synaptic strength from low-frequency tuning neurons in the left striatum is selectively potentiated (based on previous work[21]); and the difference between left and right striatal activity is the driving force for the behavioral choice (Supplementary Fig. 10A). In control condition, the driving force is A (Supplementary Fig. 10A); when the MGB inputs are inhibited, the striatal activity on both sides of the striatum decreases by dividing the same factor, B (B > 1), therefore the driving force will be A/B (Supplementary Fig. 10B). The decrease of the driving force will increase the ambiguity in making the behavioral choice, thus flatten the psychometric curve of task performance.

Together, our current work serves as a first set of explorations that both thalamic and cortical projections play roles in regulating auditory decision-making.

## Methods

**Animals**. Animal procedures were approved by the Stony Brook University Animal Care and Use Committee and carried out in accordance with National Institutes of Health standards. Male C57BL/6J mice (Charles River) were housed with free access to food, but water restricted after the start of behavioral training. In training days, water was available during task performance (2.5 μl for each correct trial); in nontraining days water bottles were provided to the mice for at least 1 h per day.

**Viral injections**. Four-week-old mice were anaesthetized with 100 mg kg$^{-1}$/ 10 mg kg$^{-1}$ ketamine/xylazine mixture and placed in a stereotaxic apparatus. Viral injections were performed using previously described procedures[21] at the following stereotaxic coordinates: Auditory striatum, 1.7 mm caudal from bregma, 3.15 mm lateral from midline, and 2.0–2.5 mm depth from cortical surface; ACx, 2.3–2.8 mm caudal from bregma, 4.15 mm lateral from midline, 0.5 mm from pia; MGB, 3.2 mm caudal from bregma, 2.0 mm lateral from midline, and 2.8 mm depth from cortical surface. A small craniotomy was made according to the coordinates, and a home-made glass micropipette (tip diameter 10–15 μm) was inserted from the surface of the brain. Virus were delivered through the glass pipettes that were connected to a Picospritzer II microinjection system (Parker Hannifin Corporation). For anterograde tracing of axons from ACx and MGB neurons, AAV2/9-CAG-dTomato (University of North Carolina Vector Core, Chapel Hill, NC) was injected into left ACx of mice, and AAV2/9-CAG-EGFP (University of North Carolina Vector Core, Chapel Hill, NC) was injected into left MGB of the same mice. For retrograde labeling of MGB neurons projecting to ACx or auditory striatum, CTB-594 or CTB-488 (0.3 μl, ThermoFisher Scientific, Waltham, MA) was injected into the left ACx and auditory striatum, respectively. Fourteen days (for AAV) or 7 days (for CTB) after injection, mice were perfused, and brain slices were collected for imaging. Images were acquired with a laser-scanning confocal microscope (FV1000, Olympus) and analyzed using FV10-Viewer (Olympus). In our analysis we defined the borders between MGd, MGv, MGm, and SG based on mouse brain atlas registration and area proportion estimation.

In our analysis we only included animals that are post hoc confirmed for adequate ArchT (or hM4Di) expression in either the primary ACx or MGB.

**Validation of inhibition capability mediated by chemo- and optogenetic tools**. We injected AAV expressing hM4Di or ArchT into MGB, and the transgene-positive axon terminals can be found in the auditory striatum. In the hM4Di validation experiment, we co-injected AAV expressing Channelrhodopsin 2 (ChR2) into the MGB of mice, and delivered blue light pulses to activate ChR2-positive axon terminals on acutely prepared striatal slices. In the ArchT validation experiment, we electrically stimulate the local axonal terminals (a bipolar electrode was placed at the border of internal capsule, and the stimulations were generated by a Stimulus Isolator (A365, World precision Instruments)), that activate both MGB and other axon terminals projecting to the auditory striatum.

In the presence of Bicuculline to exclude local GABAergic activities, we recorded the evoked excitatory postsynaptic currents (EPSCs) from striatal neurons on acutely prepared brain slices. The stimulations were adjusted to produce stable EPSCs (<100 pA) in amplitude prior to the initiation of experimental recording.

In the experiment of hM4Di, we observed a substantial reduction of evoked EPSC amplitude in the presence of CNO (10 μM in bath solution) from brain slices expressing hM4Di, but no change from brain slices without hM4Di expression (Supplementary Fig. 2). This suggests that this hM4Di-mediated inhibition of MGB neurons is sufficient to dampen the activity of the MGB projection.

In the experiment of ArchT, we activated ArchT via orange light pulses through the objective. We recorded the evoked EPSC from striatal neurons in the conditions with or without light pulses. Light pulses were started at 100 ms before electrical stimulation and lasted for 200 ms, as suggested by previous report[42]. We found that light pulses significantly suppressed evoked EPSC (Supplementary Fig. 6A). We should point out that there was a small remaining response likely due to stimulation of nonthalamic terminals and/or of terminals from non-infected thalamic neurons. Light pulses alone did not induce any significant changes on recorded striatal neurons (Supplementary Fig. 6B).

**Behavioral training**. The mice were placed on a water deprivation schedule and trained to perform an auditory 2AFC task in a single-walled sound-attenuating training chamber as described previously[20]. In brief, freely moving mice were trained to initiate a trial by poking into the center port of a three-port operant chamber, which triggered the presentation of a stimulus. Subjects then selected the left or right goal port. The cloud-of-tones stimulus consisted of a stream of 30-ms overlapping pure tones presented at 100 Hz. The stream of tones continued until the mouse withdrew from the center port. Eighteen possible tone frequencies were logarithmically spaced from 5 to 40 kHz. For each trial either the low stimulus (5–10 kHz) or high stimulus (20–40 kHz) was selected as the target stimulus, and the mice were trained to report low or high by choosing the correct side of port for water reward. Correct responses were rewarded with water (2.5 μl for each correct trial), and error trials were punished with a 4 s timeout. Sound intensity was calibrated at 65 dB. Behavioral system is control by Bpod system (Sanworks).

Stimulus strength $r$ determined the difference in the rate of high- and low-octave tone in the stimulus. Tones were drawn from the target octave with a probability of $1 + 2r/100/3$. To quantify mice's performance in the task, we used a logistic regression model described before[20]. $\log(p/(1-p)) = \beta_0 + r^*\beta_1$, where $p$ is the fraction of choices towards the port associated with high frequencies. Parameters $\beta_0$ and $\beta_1$ measure the bias and slope of the psychometric curve. Reaction time was calculated as the time between the onset of tone and the time of withdraw from the center port.

**Chemogenetic manipulation and cannula implantation**. For chemogenetic manipulation of the thalamostriatal projections, well-trained mice were bilaterally injected with the AAV2/8-hSyn-hM4Di-mCherry (Addgene, #50475) into the MGB. The surgical procedure was the same as described above in viral injection section. Stainless steel guide cannulae (26 G, 4.0 mm, Plastics One) were implanted bilaterally 2.0 mm down from the surface and were fixed to the skull with adhesive luting cement (C&B Metabond) and acrylic dental cement (Lang Dental Manufacturing, IL). A dummy cannula was inserted into each guide cannula to seal off the opening. Mice were allowed to recover from surgery for a minimum of 1 week, during which they were handled and habituated to the infusion procedure on a daily basis. After recovery, mice were back to routine daily behavioral training. Once their task performances were recovered stably (usually 4–6 weeks after the surgery), mice were infused with clozapine-n-oxide (CNO, Enzo Life Sciences, 20 μM 0.3 μl) or saline (0.3 μl) 30 min before the tested behavioral sessions. We validated the spread of CNO from cannula infusion by co-infusing a fluorescent dye (DIO, 2.5 mg/ml, 300 nl, Thermo Fisher Scientific) in a group of control mice, and identifying the DIO signals from brain slices (Supplementary Fig. 3).

**Tetrode recording**. Custom tetrode and optic fiber arrays were assembled in house. Each array carried 8 tetrodes and 1 optic fiber (62.5 μm diameter with a 50-μm core; Polymicro Technologies). Each tetrode consisted of 4 polyimide-coated nichrome wires twisted together and gold-plated to an impedance of 0.3–0.5 MΩ at 1 kHz (wire diameter 12.7 μm; Sandvik in Palm Coast, FL). The fiber tips were sharpened at the points using a diamond wheel to improve tissue penetration and increase the light illuminating area. The resulted optrodes were mounted on vertically movable microdrive. The optrode tips were coated with DiI to assist the identification of fiber tracks in brain tissues. To implant the optrode

array, mice were anaesthetized with 1–1.5% isofluorane and placed in a stereotaxic apparatus (Kopf). A craniotomy was made over the target area. The dura was removed and the implant was placed over the target area and fixed in place with dental acrylic. The tetrode was then lowered down to the auditory striatum with close recording monitoring (75 μm maximum per day).

For recording of striatal tonal responses, the mice were placed in a single-walled sound-attenuating training chamber and pure tone were played at 0.5 Hz. Tone frequencies spanned from 2 to 50 kHz and played in a random order with intensities of 50, 60, or 70 dB SPL. Electrical signals in auditory striatum were recorded using Neuralynx Cheetah 32-channel hybrid system and cheetah data acquisition software. Signals were filtered 600–6000 Hz. Single units were isolated offline using Spike3D and MClust. Because in our recordings most of the striatal neurons did not respond to tonal stimuli lower than 50 dB, we used either 60 or 70 dB (the lower responsive intensity) for tonal response analysis.

All the isolated single units having waveforms with a half-valley-width less than 100 μs are classified as FS, those having waveforms with half-valley-width more than 150 μs are classified as ChI, and all other single units with a half-valley-width between 100 and 150 μs were classified as putative MSNs.

To classify the sound response for each isolated single unit, we first used a 3 ms sliding window to obtain peristimulus time histogram (PSTH) starting from 100 ms before the tone onset to 200 ms after the tone ending. We next calculated the mean and standard deviation of the baseline firing rate using the PSTH values from 50 to 0 ms before the tone onset. Then, we determined the response time window as the period that the PSTH is more than three times standard deviation away from the mean of baseline. If there is no time point crosses this threshold, the unit is considered not responding to tones. (1) For response windows started after the tone onset and ended before 50 ms after the tone onset, we grouped them as onset response; (2) for response windows started after the tone onset and ended after 50 ms after the tone onset, we grouped them as sustained response; (3) for response windows started within 20 ms after the tone ending, we grouped them as offset response; (4) for all the response with PSTH more than two times standard deviation smaller than the mean of baseline, we grouped them as suppression response.

We averaged the increased firing rates for each frequency, and plotted the increased firing rates against corresponding frequencies. We then fitted these tuning curves with a Gaussian function:[43] $R(f) = A \times \exp(-0.5 \times (\theta - f_0)^2 / \sigma^2) + B$, where $B$ represents the baseline response, $A$ represents the amplitude of the strongest evoked response, $f_0$ represents the preferred frequency, and $\sigma$ is the tuning width. We included cells that were well fit by a Gaussian, using a criterion of $R^2 > 0.4$.

To quantify the differences of firing rates at best frequencies between control and light-on conditions, the best frequencies of individual units are separately determined by the control trials and light-on trials. The average tuning curves are calculated by centering individual tuning curves to their own best frequencies.

**Optogenetic manipulation.** For optogenetic manipulation of the thalamostriatal or corticostriatal projections, well-trained mice were bilaterally injected with the AAV9-CAG-ArchT-GFP (UNC vector core) into the MGB or the primary ACx. Laser light was adjusted to produce the desired output at the end of the patch cord. For ArchT, 530 nm laser light was generated from a diode-pumped solid-state laser (Shanghai Dream Lasers, Shanghai, China) and couple to the optic fiber through an FC/PC patch cord using a FiberPort Collimator (Thor Labs), yielding an average power of 5 mW at the optic fiber tip outside the tissue. Light pulses were started at 100 ms before sound onset and lasted for 300 ms, as suggested by previous report[42] and validated on striatal slice (detail in above Validation section). Manipulation trials were randomly interleaved with control trials.

To facilitate the penetration of optic fibers through brain tissue, we grinded the fiber tips that also scraped off the polyimide coated around the fiber tip (50–100 μm). Thus, the fiber tip surface is larger than that of untreated fiber. As suggested by previous study[44], we predicted light output in vivo using Monte Carlo modeling approach. According to our 5 mW output from 62.5 μm 0.22 NA fibers, we estimate the irradiance value as 35 mW/mm².

**Data analysis.** All data were analyzed in MATLAB.

Behavior performance analysis only included completed trials. The accuracy (percent correct) at each difficulty (evidence strength) was plotted in a psychometric plot (Fig. 1c). We calculated the evidence strength as following: (# high tones − # low tones)/ (# high tones + # low tones); −1 and 1 are the easiest, providing full evidence, and 0 is the hardest, providing no evidence for making a decision. Average performance in saline sessions (the sessions immediately before the CNO sessions) were compared with the CNO sessions from each mouse to analyze CNO effects. Population data across selected mice were used for statistical analysis. CNO effects by investigators were blinded to treatment. Experiments were repeated at least three times to ensure reproducibility, and at least three animals from each group were included in the analysis. The included animals were selected based on their completion of the experiments, the quality of recordings, and the correct locations of viral infection validated post hoc.

Studies used male mice aged 4–6 weeks at the time of viral injection and 8–14 weeks during performance of behavior tasks, tetrode recording and whole-cell recording. For analysis across animals, the comparison groups were always littermates or same purchase batch via random selection.

Only tone-responsive neurons were included in this study. Tone responses were determined by the averaged firing rates within the response windows after onset of tones. The response windows were defined for individual neurons: a 3 ms sliding window was used to calculate the averaged firing rates after the onset of tones (up to 200 ms), the period in which the firing rates are significantly (>3 standard deviation) higher than the baseline (average firing rate at 0–50 ms before the onset of tones) was used as the response window. Response windows were determined in control conditions (tones only, without light pulses). The onset of tones was aligned at 0 ms, and the first time point of the response window was defined as the response latency. Response firing rates were calculated by subtracting the baseline firing rates from the firing rates in the response windows for each neuron. The frequency of the tones that evoked the highest response firing rate was defined as the best frequency for each neuron. The response firing rates were plotted against corresponding tone frequencies for each neuron, and the resulted tuning curve was smoothened with the Savitzky-Golay method with span five. The range of frequencies at half value of the peak firing rate from the smoothened tuning curve was defined as the tuning width.

**Code availability.** The codes used in the current study are available from the corresponding author on reasonable request.

## Data availability
The datasets generated during and/or analyzed during the current study are available from the corresponding author on reasonable request.

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

## Acknowledgments

We thank Drs. Craig Evinger, Alfredo Fontanini, Yanhua Huang, Santiago Jaramillo, Mary Kritzer, Hysell Oviedo, Joshua Plotkin, Lorna Role, Yang Yang, Anthony Zador, and lab members in Ge's & Xiong's laboratories for their critical comments on this manuscript. We thank Dr. Joshua Sanders for the helps on setting up hardware and software that were used in this study. This work is supported by departmental internal funding and NIH R21DC016746 to Q.X.

## Author contributions

L.C., X.W., S.G. and Q.X. designed the experiments. L.C. and X.W. performed the experiments. L.C., X.W., and Q.X. analyzed the data and wrote the manuscript.

## Additional information

**Competing interests:** The authors declare no competing interests.

