## [Peer Review File · Nature Communications]

Reviewers' Comments:

Reviewer #1:

Remarks to the Author:

[The manuscript file contains neither page numbers nor line numbers. I don't know whether that is an omission by the authors or a problem with the journal's PDF generation software but please do not send out manuscripts without page and line numbers in future.]

Chen et al aimed to find out the relative contribution of the thalamostriatal and corticostriatal projection to striatal sound processing and frequency discrimination. They show that inhibition of the thalamostriatal projection impairs sound discrimination performance and that inhibiting the thalamostriatal projection has a slightly different effect on the response properties of the striatal neurons than inhibiting the corticostriatal projection.

The question that they address (the contribution of these two projections to the function of the striatum) is an important one. However, I have some issues with their interpretation of the data. Please see specific comments below. In places, the manuscript would benefit from more technical detail.

Introduction:

"...these inputs carry similar synaptic activities...". What does that statement mean?

"..extensively studied..." I consider that an overstatement

"...suggest that THE thalamostriatal projection..." Here and in several other places of the manuscript 'the' is missing. The manuscript is written quite well but please get a native English speaker to proofread the manuscript and iron out the remaining grammatical errors.

"Most previous studies..." reference(s)?

Results:

Please provide more details on the slice control experiments (Fig 1B). E.g. how were you selectively stimulating thalamo-striatal axons. Please also discuss the implications of the finding of Gomez et al 2017 who found that, with the CNO concentration used here CNO competitively inhibits binding at various receptor types for these experiments.

Suppl fig 2: axis label "midial" should say 'medial'.

"Our current findings on thalamocortical projections..." I believe this should read 'thalamostriatal'.

Fig 2A: The waveforms of the MSN and Ch1 neurons look very similar. Please show more than a single spike waveform from each example neuron so that readers get an idea of the variability.

"Together, our profiling revealed that all three major types of striatal neurons responded to tonal stimuli, and showed frequency preferences."

Please be more specific how many striatal neurons out of the total respond to tones. How many are

tuned? What criteria were used to assess this.

Fig 2E: Please add error bars.

Fig 2: More than 95% of the neurons in the striatum are MSNs according to the authors. However, the sample sizes for MSN, FS and ChI are given as 63, 98 and 18. Thus, the authors appear to be vastly undersampling the MSN population. Please explain. Could it be to do with the inclusion criteria? Are MSNs just not responding to sounds? Is there problem with the classification?

Legend for suppl fig 3D is missing.

Tracing of MGB projection targets (Suppl Fig 3). This, I believe, is a particularly weak part of the manuscript: According to the example micrograph (suppl fig 3B) the authors injected into primary auditory cortex. Not surprisingly (e.g. Llano and Sherman 2008; Vasquez-Lopez et al 2017 both of which should probably be cited when discussing the tracing results) the bulk of the retrogradely labelled neurons is found in the ventral division of the MGB and the amount of red-green overlap is low because the bulk of the striatum-projecting neurons are located in the non-lemniscal thalamus. Had the authors injected into a higher-order cortical area such as A2 a much larger number of labelled neurons would likely have been found in higher-order thalamic areas such as the MGBd. Consequently, the amount of overlap, i.e. double-labelled neurons, would have been much larger in that case. Therefore, the amount of information that we can take from these experiments is very limited. Because of these limitations a claim such as "These anatomical tracing studies showed that the auditory striatum received thalamic and cortical projections with different origins in MGB." is not justified. At best, the presented data indicate that there is little overlap between A1-projecting and striatum-projecting MGB neurons. Consequently, we still do not know how much overlap there is between thalamostriatal and thalamocortical neurons.

Why is the suppl fig 3B cut off at the bottom? A substantial number of green labelled neurons appears to be located in the peripeduncular nucleus (as also shown by Vasquez-Lopez et al 2017).

"Interestingly, several previous studies have reported that neurons in MGd are broadly tuned to tonal stimuli, while MGv neurons are sharply tuned (23,24)." The two cited papers refer to the cat. Please consult and cite the relevant mouse literature if available. For instance, Anderson and Linden 2011 and Vasquez-Lopez et al 2017. The former found relatively modest differences between the thalamic subdivisions while the latter, when recording (in awake mice) from thalamocortical terminals originating from higher-order auditory thalamus, found very little tone-evoked activity.

"Furthermore, the tonotopic organization is preserved in the MGv, auditory cortex (25) and in the auditory cortex projection to the striatum (17)." Again, please find and cite literature from the most relevant species, i.e. mouse, if available.

"together with our anatomical tracing, hinted that the two types of projections may likely relay distinct acoustic information to the auditory striatum in regulating striatal-dependent auditory decisions." As said before, the anatomical experiments presented in the paper do not show convincingly that the thalamostriatal and thalamocortical projections are distinct. Furthermore, we know that the auditory cortex projects to the MGB so the thalamostriatal projection could be relaying information submitted by the cortex to the thalamus. It would be worth discussing this issue.

"details in Methods and previous report (27)" I may be mistaken but as far as I understand the cited paper does not deal with ArchT so please explain its relevance.

"Interestingly, unlike silencing of thalamostriatal projections—which led to a frequency-independent decrease in responsiveness (Fig. 3D)—silencing of cortical projections preferentially decreased the responses of individual striatal units to their best frequencies (Fig. 4 C-E) and thus broadened their tuning width (Fig. 4F)." I have some concerns with the above interpretation and the data that this section relates to. First, the response reduction according to 3D is strongest at the best frequency so there quite clearly seems to be a frequency-dependent decrease in responsiveness. If there are additional analyses that allowed the authors to come to the conclusion that the decrease in responsiveness is frequency-independent, please share them with the reader. Second, I find that describing the effect shown in 4D as a broadening of the tuning curve to be somewhat misleading. The only change that is observed is a reduction of the response exclusively at the best frequency. Technically, this is consistent with a widening of the tuning curve given the definition of tuning width used here. However, there is no increase in firing at the flanks of the curve which is what most people may, at least conceptually, associate with a widening of a tuning curve. Therefore, I believe, it would be better to describe the difference in effects by saying something along the lines of 'Thalamostriatal suppression is less frequency specific than corticostriatal suppression'.

The neurons that the authors recorded from are, I assume, frequency-tuned so please state what criterion was used to assess whether they are, in fact, tuned. Please provide a few more example tuning curves.

The light intensity used here appears very high. 380mw/mm² according to my calculations. Have the authors tried lower light intensities? Might be worth mentioning in the methods what light intensities they tried (unsuccessfully) for the benefit of readers wanting to attempt similar experiments.

"We found that optical silencing of both thalamic and cortical terminals abolished the striatal sound responses (Fig. 5A-D)." Unless the authors can show that there is no sound-evoked activity in the striatum, they should use a slightly weaker statement, i.e. 'largely abolished'.

Discussion:

"The auditory striatum receives two parallel projections, one from the auditory cortex that received projection from the ventral MGB (the driver pathway)...". The part of the auditory cortex that receives mostly input from the MGBv is the primary auditory cortex. However, the spread of retrogradely labelled cells in the ACx shown in suppl fig 1B very clearly indicates that the striatum receives input from beyond the primary cortical area(s), i.e. from cortical areas that receive their main thalamic input from the non-lemniscal MGB so the statement is incorrect or, at the very least, imprecise.

".....and one from the dorsal MGB (the modulator pathway)." It also receives thalamic input from the peripeduncular nucleus (and possibly others) of the thalamus. In fact, according to their suppl fig 3D this projection may be even stronger than the one from MGBd (unfortunately the figure is cut off at the bottom and does not show the full extent of the auditory thalamus).

In other words, the striatum receives input from different auditory cortical areas and different auditory thalamic nuclei so the above statement is, at best, an oversimplification of the situation.

Given that the effects of suppressing the thalamostriatal and corticostriatal projection are different, would we also expect different effects on the behaviour when selectively manipulating the two projections? Have the authors directly compared the behavioural effects in the same species? Data on the behavioural effects of corticostriatal suppression in mice would nicely round this paper off. If those data are not available, what difference, if any, would they expect?

Reviewer #2:

Remarks to the Author:

Main Review:

Chen, Wang, Ge, and Xiong investigate the relative contributions of two auditory-related projections to the dorsal striatum, those arising from the MGB and those arising from the auditory cortex. The authors find that silencing the thalamo-striatal pathway during a frequency discrimination task led to decreased discrimination performance but, quite compellingly, no changes in behavior related to motivation, vigor, or motor engagement. The authors quantify the sound-responsiveness of dorsal striatal neurons using their responses to pure tones. The authors then use terminal silencing techniques to remove thalamic input, cortical input, or both inputs to the striatum while repeating their pure tone stimuli. The authors conclude (1) that cortical inputs provide a frequency-tuned input; (2) that thalamic inputs provide a sound-evoked but frequency-agnostic input; and (3) that together these two inputs (effectively) are the source of all sound-related activity in dorsal striatal neurons.

This is a nicely executed and nicely written set of experiments. I am particularly impressed by the clear evidence provided in SupFig 4 that their terminal suppression protocol works *ex vivo*. Having personally tried to prove that this technique works using *in vivo* measures (and failing every time), I appreciated seeing this clear evidence. The questions and concerns that I have outlined below primarily stem from the interpretation of the *in vivo* data in light of the framework laid out in the abstract. In particular, the abstract states that "In contrast to the role of the corticostriatal projection in sound frequency discrimination, the function of the auditory thalamostriatal projection is unknown". First, it is unclear to me whether the behavioral experiments presented here clarify the relative roles of these two projections for frequency discrimination. Second, it remains unclear to me whether removal of the frequency-agnostic input that the thalamus provides to the striatum (Fig. 3) ought to lead to the behavioral changes observed in Fig. 1.

Major Concerns:

1. It is unclear how silencing the non-frequency-specific input from MG to striatum causes the behavioral deficits observed in Fig. 1. I am not claiming that the behavior and physiology data are inconsistent with one another, simply that I can't tell whether they are or not. It is not intuitively clear to me how Fig. 3 leads to Fig. 1. For example, from Fig. 5E, the authors think that the MG-striatum projection is providing a non-specific but sound-driven input, and that when removing this input, neurons retain their BF, BW, etc. It seems like all the information is there to still do a good job at this task. At minimum the authors should elaborate on this and point to references that have seen this type of change in behavior in the past by removing a tonic component of a tuning curve (probably in other system). Perhaps more compellingly, a simple model that translates striatal activity to behavior choice would be fantastic if it could show how removing the non-frequency-specific input would lead to a flattening of choice behavior.

2. In the paper that introduced the cloud of tones experiment, the authors unilaterally silenced the cortical input to the striatum (in rats) and showed rightward and leftward shifts in the psychometric functions. I am not aware of anyone bilaterally silencing the cortical projection and observing changes in behavior (perhaps I'm wrong, but I don't see it referenced here either). My point is that we don't know what the manipulation performed here, when applied to the cortical-striatal pathway, would do to behavior. Therefore it seems disingenuous to claim that we already knew how the cortical-striatal pathway contributes to behavior, that we now aim to study the thalamo-striatal pathway, and then say it's now figured out. This seems like a comparison of apples to oranges.

The most satisfying solution to this would be to perform the bilateral silencing of the cortical-striatal pathway. Does this cause a change in behavior different than that observed in the figures presented herein? One might hope so considering the authors have gone to great lengths to show that cortex and thalamus convey different information to the striatum. Then, could these differences in behavior be explained by the differences in the acoustic information that is being provided by these two inputs. Knowing this would take us a long way toward understanding how cortical and thalamic inputs differentially contribute to striatum-dependent frequency discrimination behavior. In lieu of that, it seems necessary to discuss in detail the differences between studies that looked at silencing the cortico-striatal pathway during behavior and those performed here.

3. The authors do a nice job showing that thalamic projections to the striatum and cortex arise largely from different MG subdivisions. It would be very helpful to know from where within the auditory cortex are these projections to dorsal striatum arising? Looking at Supp. Fig. 1 it's hard to tell where the boundaries between dorsal, ventral and primary auditory cortex are.

4. Fig. 5E is a nice and simplified version of how these 2 projections can account for the sound-responsiveness observed the dorsal striatum. However, based on Figs 3 and 4, I would like more quantification on the degree to which this is true. For example, in Fig. 3, can the "laser" tuning curves really be modeled as the "control" tuning curves minus some offset? I can't tell whether gain control or offset is a better model for how thalamic input is contributing to MSN spiking. In general I would appreciate more quantification. One nice quantification would be fitting the line that transforms the control to the laser tuning curves. What are the slope and offset values of these lines for each cell?

5. The abstract says that the thalamic input mainly acts as a gain controller. That does not seem to be supported by the data here. A gain controller would be divisive, and it seems to me that the authors conclude that the thalamic input is providing an offset.

6. For figures 3,4,5, quantify and show the best frequency as a function of laser on/laser off conditions (for those neurons that still have a best frequency).

7. In Fig. 5, show the equivalent of panels E,F from Figs. 3,4. (The 9 is much smaller here, which is all the more reason to show all of the data points).

8. Why not report here the changes in tuning curves also for putative FS and ChI cells? Perhaps the n's are very low?

Minor comments and concerns:

1. Under tetrode recordings: "Each tetrode is consisted of 4 polyimide-coated nichrome twisted..." The word "is" can be removed and I believe the word "wire" should be inserted between "nichrome" and "twisted".

2. It seems that a more accurate title would be that thalamic and cortical projections differentially "contribute to" striatal sound representations, rather than control.

Reviewer #3:

Remarks to the Author:

In the manuscript Liang Chen and colleagues use a combination of behaviour, pathway specific

chemo-genetics and in vivo electrophysiology to demonstrate that the thalamo-striatal projections from the dorsal portion of the medial geniculate nucleus to the striatum is critical for performance on an auditory discrimination task. They go on to show that the cortical and thalamic inputs to the auditory striatum, play a different role in shaping the striatal sound responses. The writing is clear and their conclusions are supported by their data. I just have one main concern that may affect the interpretation of the results, otherwise I think the manuscript offers a very interesting contribution to the field and paves the way for more investigation of the thalamo-striatal projection which has been neglected.

My main concern is to do with potential discrepancies in the amount of the auditory cortex and MGd that will be infected with the injections of the AAV-ArchT. Presumably a large proportion of the MGd is covered in their ArchT injections, whether the whole auditory cortex is covered is not clear. This could have consequences for the interpretation of their results as if a smaller proportion of the auditory cortex is infected this could lead to an underestimate of the cortical contribution to auditory responses in the striatum. In addition, could not covering the whole of the auditory cortex also potential lead to observations that the peak response is preferential suppressed? To start to address this the authors could at least show the spread and cortical coverage of their ArchT injections. In addition, if the injections did not cover the entire cortical area I would suggest that a few animals are added to the study where a large cortical injections covering all if not more than the auditory cortex are performed.

I think it would also be beneficial to show how the different MSN response types (sustained, onset, offset and suppression) are effected by the inhibition. It is not clear from the bulk analysis if the tuning of each response class is equally affected by the optical inhibition. This is important as in the individual examples different response types are shown.

Minor points include,

Figure 1. Could the authors check if the images have been reversed as the red image looks like a cortical injection as the axonal tracks characteristic of cortical fibres passing through the striatum are present. This may also be a feature of MGd-striatal projections but I wanted to check.

Figure 3,4,5. In panel A for all figures AchT should be replaced with ArchT.

Page 4 paragraph 3, it would be helpful to include how the electrical stimulation is performed in the main text. i.e local electrical stimulation within the striatum.

Page 7 paragraph 3, the authors mention that the offset and suppression MSN response types are generally not frequency tuned, yet the two example neurons appear to have a frequency preference. Could this be quantified for all responses types in a supplemental figure?

Overall the manuscript is clear and the authors summary of modulatory and driver pathways is very interesting.

We thank the reviewers and editor for their careful reading and constructive remarks.
We have taken the suggestions and revised the manuscript accordingly. Please find
below a detailed point-by-point response to all comments (reviewer's comments in grey
and italic, our responses in black).

**Reviewer #1**

*[The manuscript file contains neither page numbers nor line numbers. I don't know whether that is an*
*omission by the authors or a problem with the journal's PDF generation software but please do not send*
*out manuscripts without page and line numbers in future.]*

We have added page numbers and line numbers to the manuscript.

*Chen et al aimed to find out the relative contribution of the thalamostriatal and corticostriatal projection to*
*striatal sound processing and frequency discrimination. They show that inhibition of the thalamostriatal*
*projection impairs sound discrimination performance and that inhibiting the thalamostriatal projection has*
*a slightly different effect on the response properties of the striatal neurons than inhibiting the*
*corticostriatal projection.*

*The question that they address (the contribution of these two projections to the function of the striatum) is*
*an important one. However, I have some issues with their interpretation of the data. Please see specific*
*comments below. In places, the manuscript would benefit from more technical detail.*

We thank the reviewer's comments on the importance of this work.

*1. "...these inputs carry similar synaptic activities...". What does that statement mean?*

We have revised and clarified the statement in the lines of 6-9 on Page 3. "... The axon
terminals of cortical and thalamic projections converge with comparable densities onto
individual striatal neurons, forming functional glutamatergic synapses (i.e.
thalamostriatal and corticostriatal synapses)."

*2. "...extensively studied..." I consider that an overstatement*

We revised the statement as now at Page 3 lines 9-10: "How these two projections
coordinate to regulate the striatal activities and striatal-dependent behaviors remain
largely unknown."

*3. "...suggest that THE thalamostriatal projection..." Here and in several other places of the manuscript*
*'the' is missing. The manuscript is written quite well but please get a native English speaker to proofread*
*the manuscript and iron out the remaining grammatical errors.*

We thank the reviewer for pointing out these errors. We have worked hard to correct
these errors across the manuscript, and have engaged additional expertise to improve
the manuscript in terms of language and grammar.

*4. "Most previous studies..." reference(s)?*

We have added references (Page 3 line 14).

*5. Please provide more details on the slice control experiments (Fig 1B). E.g. how were you selectively*
*stimulating thalamo-striatal axons. Please also discuss the implications of the finding of Gomez et al 2017*
*who found that, with the CNO concentration used here CNO competitively inhibits binding at various*
*receptor types for these experiments.*

We have performed additional experiments to validate this CNO/hM4Di-mediated
terminal inhibition method as presented in **Supplementary Fig. 2 A&B**. To achieve the
selective stimulation of thalamostriatal axons, we co-injected AAV-hM4Di-mCherry and
AAV-ChR2 into the MGB of mice, and performed whole-cell patch recordings on acutely

prepared striatal slices. We delivered blue light pulses through 40X objective to focally
activate ChR2-positive axon terminals around the recording sites. We have revised the
related description in the main text (page 4 lines 20-23) and **Methods**.

We agree that any experiments with either opto- or chemogenetic method needs a
careful calibration to avoid any confounding effect from nonspecific stimulation. In these
slice recordings using the current CNO dose, we observed substantial/specific inhibition
of synaptic transmission from preparations with hM4Di expression but not in control
preparations (i.e. no hM4Di expression), shown in **Supplementary Fig. 2 A&B**.
Furthermore, to avoid any other confounding effects from non-neuronal activity effect, in
the behavioral experiments, we have included a group of control mice that did not
express hM4Di (**Fig. 1D-H**, mCherry alone group). We did not observe any change
upon CNO infusion as compared to the saline infusion sessions, indicating that CNO
has no significant side effects on the behavior we are testing.

We agree with the reviewer about this concern and have added a short description
about the testing for this side effect raised in the study by (Gomez, Bonaventura et al.
2017) (page 5 line 3-7).

*6. Suppl fig 2: axis label "midial" should say 'medial'.*

We have corrected this typo.

*7. "Our current findings on thalamocortical projections..." I believe this should read 'thalamostriatal'.*

We have corrected this typo.

*8. Fig 2A: The waveforms of the MSN and Chl neurons look very similar. Please show more than a single
spike waveform from each example neuron so that readers get an idea of the variability.*

We agree that it was difficult to visualize these differences. We have now revised **Fig.**
**2A** by overlaying the waveforms of the three cell types to help readers to see the
differences in the waveforms.

The main difference in waveforms is that Chl neurons have a longer trough (valley)
duration than do MSN neurons. To clarify this point, we now detail waveform criterion in
the **Methods** "All the isolated single units having waveforms with a half-valley-width less
than 100 μ s are classified as fast-spiking interneurons (FS), those having waveforms
with half-valley-width more than 150 μ s are classified as cholinergic interneurons (Chl),
and all other single units with a half-valley-width between 100 and 150 μ S were
classified as putative MSNs". (Page 19 lines 21-22 & page 20 lines 1-2)

*9. "Together, our profiling revealed that all three major types of striatal neurons responded to tonal stimuli,
and showed frequency preferences." Please be more specific how many striatal neurons out of the total
respond to tones. How many are tuned? What criteria were used to assess this.*

We have now included all these numbers in the main text (page 7 lines 9-15 & 22-23).
In brief, we identified 206 of 1178 neurons responding to tonal stimuli (MSN:70/944;
FS:118/152; Chl:18/82). Out of the 206, we found 160 were tuned to pure tones. We
included the tuned neurons in the tuning analysis.

We have included the criteria for defining tone-responsive neurons and tuning in the
**Methods**. "To classify the sound response for each isolated single unit, we first used a
3 ms sliding window to obtain a peristimulus time histogram (PSTH) starting from 100
47 ms before the tone onset to 200 ms after the tone offset. We next calculated the mean
and standard deviation of the baseline firing rate using the PSTH values from 50 to 0

1 ms before the tone onset. Then, we determined the response time window as the period that the PSTH is more than 3 times the standard deviation away from the mean of baseline. If no time point crosses this threshold, the unit is considered not responding to the tones...”

To determine whether a unit is tuned to pure tones, “we averaged the increased firing rates for each frequency, and plotted the increase in firing rate against corresponding frequencies. We then fitted these tuning curves with a Gaussian function (Moore and Wehr 2013): $R(f) = A \times \exp(-0.5 \times (\theta - f_0)^2 / \sigma^2) + B$, where B represents the baseline response, A represents the amplitude of the strongest evoked response, f_0 represents the preferred frequency, and σ is the tuning width. We included cells that were well fit by a Gaussian, using a criterion of $R^2 > 0.4$.” (Page 20 lines 3-20)

10. Fig 2E: Please add error bars.

We have added error bars in the **Fig. 2E**.

11. Fig 2: More than 95% of the neurons in the striatum are MSNs according to the authors. However, the sample sizes for MSN, FS and Chl are given as 63, 98 and 18. Thus, the authors appear to be vastly undersampling the MSN population. Please explain. Could it be to do with the inclusion criteria? Are MSNs just not responding to sounds? Is there problem with the classification?

The finding that ~95% neurons in the striatum are MSNs is based on nuclear morphology analyses (Graveland and DiFiglia 1985). Sampling from tetrode recordings may be influenced by firing properties of individual neuronal types. In our tetrode recorded population, 80.1% (944/1178) neurons were classified as MSNs, 12.9 % (152/1178) were FSs, and 7% (82/1178) were identified as Chls. This closely resembles the distribution of neuronal types reported in previous studies (Berke 2008, Gage, Stoetzner et al. 2010). Adjusting the classification criteria caused little change in the distribution of cell types. We have included the quantification in our main text (page 7 lines 9-15).

Among our identified neurons, 7.4% (70/944) MSNs, 77.6% (118/152) FSs, 22% (18/82) Chls showed tone responses. Indeed, there was a smaller percentage of MSNs responding to tonal stimulation. This may be due to unique circuitry or cellular mechanisms, which are interesting questions to explore in the future. We also added a short discussion on this point (page 13 lines 16-20).

12. Legend for suppl fig 3D is missing.

We have corrected this oversight and have now added the legend.

13. Tracing of MGB projection targets (Suppl Fig 3). This, I believe, is a particularly weak part of the manuscript: According to the example micrograph (suppl fig 3B) the authors injected into primary auditory cortex. Not surprisingly (e.g. Llano and Sherman 2008; Vasquez-Lopez et al 2017 both of which should probably be cited when discussing the tracing results) the bulk of the retrogradely labelled neurons is found in the ventral division of the MGB and the amount of red-green overlap is low because the bulk of the striatum-projecting neurons are located in the non-lemniscal thalamus. Had the authors injected into a higher-order cortical area such as A2 a much larger number of labelled neurons would likely have been found in higher-order thalamic areas such as the MGBd. Consequently, the amount of overlap, i.e. double-labelled neurons, would have been much larger in that case. Therefore, the amount of information that we can take from these experiments is very limited. Because of these limitations a claim such as “These anatomical tracing studies showed that the auditory striatum received thalamic and cortical projections with different origins in MGB.” is not justified. At best, the presented data indicate that there is

1 *little overlap between A1-projecting and striatum-projecting MGB neurons. Consequently, we still do not*
*know how much overlap there is between thalamostriatal and thalamocortical neurons.*

We fully agree with the reviewer about the limits in interpretation of the tracing
experiments, which was resulted from an ambiguous statement of the focus of this
manuscript. We have now revised the entire manuscript to clarify that in current study,
we focused on how the projections from the primary auditory cortex and MGB regulate
the auditory striatal sound representation and striatal-related frequency discrimination
behavior. We have added suggested references in the related discussion.

The auditory striatum indeed may receive functional inputs from the secondary auditory
cortex and other thalamic nuclei. However, as suggested in our results (**Fig. 5 B-E**) the
tone-evoked striatal responses were largely abolished by silencing both projections from
the primary auditory cortex and MGB, indicating that these two pathways are dominant
in controlling the striatal tone responses in our current study. We agree that our study
does not exclude the involvement of other thalamic projections, which are very
interesting to us and will be investigated in the future.

We have discussed these points in the **Discussion** section (Page 12 lines 21-23, page
13 lines 1-2).

*14. Why is the suppl fig 3B cut off at the bottom? A substantial number of green labelled neurons appears*
*to be located in the peripeduncular nucleus (as also shown by Vasquez-Lopez et al 2017).*

We revised the **Suppl Fig. 5B** (was Supple. Fig. 3B). Indeed, our tracing results also
found that peripeduncular nucleus projects to the auditory striatum, consistent with the
finding of Vasquez-Lopez et al 2017. As stated in response to Q13 above, we now
discussed the peripeduncular nucleus together with other projections. These projections
will be interesting targets for future exploration.

*15. "Interestingly, several previous studies have reported that neurons in MGd are broadly tuned to tonal*
*stimuli, while MGv neurons are sharply tuned (23,24)." The two cited papers refer to the cat. Please*
*consult and cite the relevant mouse literature if available. For instance, Anderson and Linden 2011 and*
*Vasquez-Lopez et al 2017. The former found relatively modest differences between the thalamic*
*subdivisions while the latter, when recording (in awake mice) from thalamocortical terminals originating*
*from higher-order auditory thalamus, found very little tone-evoked activity. "Furthermore, the tonotopic*
*organization is preserved in the MGv, auditory cortex (25) and in the auditory cortex projection to the*
*striatum (17)." Again, please find and cite literature from the most relevant species, i.e. mouse, if available.*

We have included these references.

*16. "together with our anatomical tracing, hinted that the two types of projections may likely relay distinct*
*acoustic information to the auditory striatum in regulating striatal-dependent auditory decisions." As said*
*before, the anatomical experiments presented in the paper do not show convincingly that the*
*thalamostriatal and thalamocortical projections are distinct. Furthermore, we know that the auditory cortex*
*projects to the MGB so the thalamostriatal projection could be relaying information submitted by the*
*cortex to the thalamus. It would be worth discussing this issue.*

As indicated in the response in Q13, we revised the entire manuscript to clarify that the
current study focuses on the projections from the primary auditory cortex and MGB to
the auditory striatum. We also stated in the discussion "...The modulatory pathway
refers to the inputs that the dorsal and medial MGB receive from both the inferior
colliculus and the primary auditory cortex..." (page 12 lines 6-8).

17. “details in Methods and previous report (27)” I may be mistaken but as far as I understand the cited
paper does not deal with ArchT so please explain its relevance.

In the cited paper, they used eArch3.0 which is an analog of ArchT. Inspired by their
findings, we set up our silencing protocol using ArchT, and validated *ex vivo* as shown
in **Supplementary Fig 6**. We revised the statement in the main text and **Methods** to
make this clear.

18. “Interestingly, unlike silencing of thalamostriatal projections—which led to a frequency-independent
decrease in responsiveness (Fig. 3D)—silencing of cortical projections preferentially decreased the
responses of individual striatal units to their best frequencies (Fig. 4 C-E) and thus broadened their tuning
width (Fig. 4F).” I have some concerns with the above interpretation and the data that this section relates
to. First, the response reduction according to 3D is strongest at the best frequency so there quite clearly
seems to be a frequency-dependent decrease in responsiveness. If there are additional analyses that
allowed the authors to come to the conclusion that the decrease in responsiveness is frequency-
independent, please share them with the reader. Second, I find that describing the effect shown in 4D as
a broadening of the tuning curve to be somewhat misleading. The only change that is observed is a
reduction of the response exclusively at the best frequency.

Technically, this is consistent with a widening of the tuning curve given the definition of tuning width used
here. However, there is no increase in firing at the flanks of the curve which is what most people may, at
least conceptually, associate with a widening of a tuning curve. Therefore, I believe, it would be better to
describe the difference in effects by saying something along the lines of ‘Thalamostriatal suppression is
less frequency specific than corticostriatal suppression’.

We agree with these comments. We revised the figures and the text to address these
concerns: 1) We added an additional analysis to characterize the effect of silencing
MGB projection to the auditory striatum (**Fig. 3H**). We found that the change in the
tuning curve caused by silencing the MGB projection can be explained as a scale down
(division) with a small offset (subtraction). Therefore, the MGB projection mainly
functions as a gain controller, with a same scale factor across broad frequencies. We
revised the corresponding conclusions in the text. 2) We agree with the reviewer that
the calculation of tuning width was misleading. We replaced this analysis with the
quantification of firing rates at $\pm 1/2$ octave frequencies and compared them between the
conditions with or without projections from the primary auditory cortex (**Fig. 4F**).

19 The neurons that the authors recorded from are, I assume, frequency-tuned so please state what
criterion was used to assess whether they are, in fact, tuned. Please provide a few more example tuning
curves.

We have included the tuning criterion in the **Methods** (also in response to Q9 above:
“We averaged the increased firing rates for each frequency, and plotted the increased
firing rates against corresponding frequencies. We then fitted these tuning curves with a
Gaussian function (Moore and Wehr 2013): $R(f) = A \times \exp(-0.5 \times (\theta - f_0)^2 / \sigma^2) + B$,
where B represents the baseline response, A represents the amplitude of the strongest
evoked response, f_0 represents the preferred frequency, and σ is the tuning width. We
included cells that were fit by a Gaussian with $R^2 > 0.4$.”). We now included more
example tuning curves in **Fig 2E**.

20. The light intensity used here appears very high. 380mw/mm2 according to my calculations. Have the
authors tried lower light intensities? Might be worth mentioning in the methods what light intensities they
tried (unsuccessfully) for the benefit of readers wanting to attempt similar experiments.

We appreciate the reviewer’s suggestion, and have included the estimation of light
irradiance values in **Methods** (page 21 lines 8-12).

Indeed, in the context of brain slice, using the theoretically predicted irradiance value
using Kubelka–Munk model ([https://web.stanford.edu/group/dlab/cgi-
bin/graph/chart.php](https://web.stanford.edu/group/dlab/cgi-bin/graph/chart.php)), the estimated light intensity will be ~ 380 mw/mm². However, to
facilitate the penetration of optic fibers through brain tissue, before implantation we
sharpened the fiber tips, which also scraped off the polyimide coated around the fiber tip
(50-100 μm). Thus, the fiber tip surface was larger than that of untreated fibers. As
suggested by previous studies using Monte Carlo modeling (Spellman, Rigotti et al.
2015, Stujenske, Spellman et al. 2015), the predicted light output *in vivo* approach with
a 5mW output from 62.5 μm 0.22 NA fibers is ~35 mW/mm².
More importantly, we tested different light intensities for the effects of ArchT-mediated
terminal inhibition on brain slices, and chose the lowest reliable effective light intensity.
The *in vivo* light setting was then determined to deliver the comparable light intensity
based on the above calculation.

21. *“We found that optical silencing of both thalamic and cortical terminals abolished the striatal sound
responses (Fig. 5A-D).” Unless the authors can show that there is no sound-evoked activity in the
striatum, they should use a slightly weaker statement, i.e. ‘largely abolished’.*

We revised the statement to “largely abolished”.

22. *“The auditory striatum receives two parallel projections, one from the auditory cortex that received
projection from the ventral MGB (the driver pathway)...”. The part of the auditory cortex that receives
mostly input from the MGBv is the primary auditory cortex. However, the spread of retrogradely labelled
cells in the ACx shown in suppl fig 1B very clearly indicates that the striatum receives input from beyond
the primary cortical area(s), i.e. from cortical areas that receive their main thalamic input from the non-
lemniscal MGB so the statement is incorrect or, at the very least, imprecise.*

We agree that this was an imprecise statement. We rephrased the discussion to clarify
that in this study we examined the projection from the primary auditory cortex (which is
in the driver pathway) and the projection from the MGB (which is in the modulator
pathway). We also discussed the projections from the secondary auditory cortex and
other auditory thalamic nuclei that need further investigation. (Page 12 lines 21-23,
page 13 line 1-2).

23. *“.....and one from the dorsal MGB (the modulator pathway).” It also receives thalamic input from the
peripeduncular nucleus (and possibly others) of the thalamus. In fact, according to their suppl fig 3D this
projection may be even stronger than the one from MGBd (unfortunately the figure is cut off at the bottom
and does not show the full extent of the auditory thalamus). In other words, the striatum receives input
from different auditory cortical areas and different auditory thalamic nuclei so the above statement is, at
best, an oversimplification of the situation.*

We agree that this was an imprecise statement. We updated the **Suppl. Fig. 5B** image
to show the complete labeling in peripeduncular nucleus. Same as the response for
Q22, we rephrased the discussion to clarify that we examined the projection from the
primary auditory cortex (which is in the driver pathway) and the projection from the MGB
(which is in the modulator pathway) in this study. We also now discuss the projections
from the secondary auditory cortex and other auditory thalamic nuclei that need further
investigation.

24. *Given that the effects of suppressing the thalamostriatal and corticostriatal projection are different,
would we also expect different effects on the behaviour when selectively manipulating the two projections?
Have the authors directly compared the behavioural effects in the same species? Data on the behavioural*

*effects of corticostriatal suppression in mice would nicely round this paper of. If those data are not*
*available, what difference, if any, would they expect?*

As suggested by the reviewer, we performed the bilateral corticostriatal inhibition in
mice and analyzed the effects on task performance using the same CNO-hM4Di
strategy as we did for thalamostriatal pathway. We observed a similar change in
psychometric curve of task performance (**Suppl. Fig. 4 A&B**) compared to what we
observed from thalamostriatal pathway silencing.

We proposed a working model by combining our previous findings(Zemanick, Strick et
al. 1991, Xiong, Znamenskiy et al. 2015) and current study (**Suppl. Fig. 10**). Both MGB
and the primary auditory cortex contribute to the tone-evoked striatal activity. In one of
our previous works, we showed that the “cloud-of-tones” task training induced selective
synaptic plasticity on corticostriatal synapses, and there is a synaptic strength gradient
across tonotopic axis in the auditory striatum from well-trained animals (Xiong,
Znamenskiy et al. 2015). In that study, we proposed a simplified model ((Xiong,
Znamenskiy et al. 2015), Suppl. Fig. 8) that translates striatal activity to behavior choice:
in well-trained animals that learned to associate low-frequency tones with rightward
choice, the synaptic strength from low-frequency tuning neurons in the left striatum is
selectively potentiated; and the difference between left and right striatal activity is the
driving force for the behavioral choice (**Suppl. Fig. 10A**).

Using the same model, we implemented our MGB inhibition finding: in control condition,
the driving force is A (**Suppl. Fig. 10A**); when the MGB inputs are inhibited, the striatal
activity on both sides of the striatum decreases by dividing the same factor, B ($B > 1$),
therefore the driving force will be A/B (**Suppl. Fig. 10B**). The decrease of the driving
force will increase the ambiguity in making the behavioral choice, thus flatten the
psychometric curve of task performance.

In the case of inhibiting the primary ACx, the tuning information in striatum is reduced.
Thus, the performance accuracy is reduced and the psychometric curve is flattened as
well.

**Reviewer #2**

*Main Review:*

*Chen, Wang, Ge, and Xiong investigate the relative contributions of to MSN firing of 2 auditory-related*
*projections to the dorsal striatum, those arising from the MGB and those arising from the auditory cortex.*

*The authors find that that silencing the thalamo-striatal pathway during a frequency discrimination task led*
*to decreased discrimination performance but, quite compellingly, no changes in behavior related to*
*motivation, vigor, or motor engagement. The authors quantify the sound-responsiveness of dorsal striatal*
*neurons using their responses to pure tones. The authors then use terminal silencing techniques to*
*remove thalamic input, cortical input, or both inputs to the striatum while repeating their pure tone stimuli.*
*The authors conclude (1) that cortical inputs provide a frequency-tuned input; (2) that thalamic inputs*
*provide a sound-evoked but frequency-agnostic input; and (3) that together these 2 inputs (effectively) are*
*the source of all sound-related activity in dorsal striatal neurons.*

*This is a nicely executed and nicely written set of experiments. I am particularly impressed by the clear*
*evidence provided in SupFig 4 that their terminal suppression protocol works ex vivo. Having personally*
*tried to prove that this technique works using in vivo measures (and failing every time), I appreciated*
*seeing this clear evidence. The questions and concerns that I have outlined below primarily stem from the*
*interpretation of the in vivo data in light of the framework laid out in the abstract. In particular, the abstract*
*states that “In contrast to the role of the corticostriatal projection in sound frequency discrimination, the*
*function of the auditory thalamostriatal projection is unknown”. First, it is unclear to me whether the*
*behavioral experiments presented here clarify the relative roles of these two projections for frequency*

*discrimination. Second, it remains unclear to me whether removal of the frequency-agnostic input that the*
*thalamus provides to the striatum (Fig. 3) ought to lead to the behavioral changes observed in Fig. 1.*

We thank the reviewer for the appreciation in experimental setup and writing. We
responded these two concerns below.

*1. It is unclear how silencing the non-frequency-specific input from MG to striatum causes the behavioral*
*deficits observed in Fig. 1. I am not claiming that the behavior and physiology data are inconsistent with*
*one another, simply that I can't tell whether they are or not. It is not intuitively clear to me how Fig. 3 leads*
*to Fig. 1. For example, from Fig. 5E, the authors think that the MG-striatum projection is providing a non-*
*specific but sound-driven input, and that when removing this input, neurons retain their BF, BW, etc. It*
*seems like all the information is there to still do a good job at this task. At minimum the authors should*
*elaborate on this and point to references that have seen this type of change in behavior in the past by*
*removing a tonic component of a tuning curve (probably in other system). Perhaps more compellingly, a*
*simple model that translates striatal activity to behavior choice would be fantastic if it could show how*
*removing the non-frequency-specific input would lead to a flattening of choice behavior.*

We agree that this section of the manuscript has ambiguity, and we have included a
simplified model to interpret how striatal activity changes would affect the behavioral
choice (**Supplementary Fig. 10**).

In one of our previous studies, we showed that the “cloud-of-tones” task training
induced selective synaptic plasticity on corticostriatal synapses creating a synaptic
strength gradient across the tonotopic axis in the auditory striatum in well-trained
animals (Xiong, Znamenskiy et al. 2015). In that study, we proposed a simplified model
((Xiong, Znamenskiy et al. 2015), Suppl. Fig. 8) that translates striatal activity to
behavioral choice. In well-trained animals that learned to associate low-frequency tones
with rightward choice, the synaptic strength from low-frequency tuning neurons in the
left striatum are selectively potentiated so that the difference between left and right
striatal activity is the driving force for the behavioral choice (**Suppl. Fig. 10A**). Using the
same model, we implemented our current finding. In the control condition, the driving
force is A (**Suppl. Fig. 10A**). When the MGB inputs are inhibited, the striatal activity on
both sides of the striatum decreases by the same factor, B ($B > 1$). Therefore, the driving
force will be A/B (**Suppl. Fig. 10B**). The decrease in the driving force increases the
ambiguity in making the behavioral choice, which flattens the psychometric curve of task
performance.

*2. In the paper that introduced the cloud of tones experiment, the authors unilaterally silenced the cortical*
*input to the striatum (in rats) and showed rightward and leftward shifts in the psychometric functions. I am*
*not aware of anyone bilaterally silencing the cortical projection and observing changes in behavior*
*(perhaps I'm wrong, but I don't see it referenced here either). My point is that we don't know what the*
*manipulation performed here, when applied to the cortical-striatal pathway, would do to behavior.*
*Therefore it seems disingenuous to claim that we already knew how the cortical-striatal pathway*
*contributes to behavior, that we now aim to study the thalamo-striatal pathway, and then say it's now*
*figured out. This seems like a comparison of apples to oranges.*

*The most satisfying solution to this would be to perform the bilateral silencing of the cortical-striatal*
*pathway. Does this cause a change in behavior different than that observed in the figures presented*
*herein? One might hope so considering the authors have gone to great lengths to show that cortex and*
*thalamus convey different information to the striatum. Then, could these differences in behavior be*
*explained by the differences in the acoustic information that is being provided by these two inputs.*
*Knowing this would take us a long way toward understanding how cortical and thalamic inputs*
*differentially contribute to striatum-dependent frequency discrimination behavior. In lieu of that, it seems*
*necessary to discuss in detail the differences between studies that looked at silencing the cortico-striatal*
*pathway during behavior and those performed here.*

In response to the suggestion, we performed the bilateral corticostriatal inhibition in
mice and analyzed the effects on task performance using the same CNO-hM4Di
strategy as we performed in the thalamostriatal pathway. We observed a similar change
in psychometric curve of task performance (**Suppl. Fig. 4A&B**) compared to what we
observed with thalamostriatal pathway silencing. As we proposed in the working model
(**Suppl. Fig. 10**), both MGB and the primary auditory cortex contribute to the tone-
evoked striatal activity. The difference between left and right sides of the striatal activity
is the driving force that leads to the behavioral decisions. Therefore, silencing either the
MGB or the primary auditory cortex will decrease the driving force and thereby reduce
performance accuracy.

*3. The authors do a nice job showing that thalamic projections to the striatum and cortex arise largely*
*from different MG subdivisions. It would be very helpful to know from where within the auditory cortex are*
*these projections to dorsal striatum arising? Looking at Supp. Fig. 1 it's hard to tell where the boundaries*
*between dorsal, ventral and primary auditory cortex are.*

We have revised the **Suppl. Fig. 1** to show the boundaries between dorsal, and primary
auditory cortex. Both the primary and secondary auditory cortices project to the auditory
striatum. In our silencing experiments, our viral injections preferentially targeted the
primary auditory cortex (**Suppl. Fig. 7**). We have revised the entire manuscript to clarify
that what we analyzed are the projections from the primary auditory cortex and MGB.

*4. Fig. 5E is a nice and simplified version of how these 2 projections can account for the sound-*
*responsiveness observed the dorsal striatum. However, based on Figs 3 and 4, I would like more*
*quantification on the degree to which this is true. For example, in Fig. 3, can the "laser" tuning curves*
*really be modeled as the "control" tuning curves minus some offset? I can't tell whether gain control or*
*offset is a better model for how thalamic input is contributing to MSN spiking. In general I would*
*appreciate more quantification. One nice quantification would be fitting the line that transforms the control*
*to the laser tuning curves. What are the slope and offset values of these lines for each cell?*

We fully agree with these suggestions. We performed the fitting (Gaussian fit to the
tuning curves) and found that the control tuning curves can be transformed to the
averaged "laser" tuning curves by a small offset (0.01 ± 0.02) and a simple division
(factor: 1.8 ± 0.2) (**Fig. 3H**). Accordingly, we revised the model in **Figure 5F** to show the
gain controller function of MGB.

*5. The abstract says that the thalamic input mainly acts as a gain controller. That does not seem to be*
*supported by the data here. A gain controller would be divisive, and it seems to me that the authors*
*conclude that the thalamic input is providing an offset.*

As we addressed in Q4 above, we showed that the effect of silencing MGB input is
mainly divisive (**Fig. 3H**).

*6. For figures 3,4,5, quantify and show the best frequency as a function of laser on/laser off conditions*
*(for those neurons that still have a best frequency).*

We have added the suggested quantification in **Fig. 3G**. However, many neurons no
longer have clear best frequencies in cases of silencing the cortical projection alone or
cortical and MGB projections together. Therefore, we didn't perform the same plots in
**Figures 4&5**.

7. In Fig. 5, show the equivalent of panels E,F from Figs. 3,4. (The 9 is much smaller here, which is all the
more reason to show all of the data points).

We have added the equivalent panel E in **Figure 5**. For the panel F in figure 3 that
quantify the tuning width, as pointed out by reviewer #1 that was misleading in the case
of cortical pathway inhibition (comment #18 2nd concern), we removed it from **Figure 4**
and do not perform the plot in **Figure 5** either.

8. Why not report here the changes in tuning curves also for putative FS and Chl cells? Perhaps the n's
are very low?

We have a total of 75 units (40 for cortical silencing and 35 for MGB silencing) that are
putative FS neurons. The tuning curve effects are shown below. The effects are very
similar to what we observed in MSN. However, we only have 7 units (4 for cortical
silencing and 3 for MGB silencing) that are putative Chl neurons, which is not sufficient
for quantification or a solid conclusion. Therefore, we only reported the projecting
neuron type, the MSNs, in the manuscript.

Effects of silencing the projections from MGB or the primary auditory cortex (ACx) to striatal FS neurons on tone-evoked responses. A. Tuning curves in the conditions with (red) or without (black) light pulses. ArchT was expressed in MGB. n=35. B. Tuning curves in the conditions with (red) or without (black) light pulses. ArchT was expressed in the primary auditory cortex. n=40. Data are presented as mean and standard error. *, $p < 0.01$, paired t test.

*Minor comments and concerns:*

1. Under tetrode recordings: “Each tetrode is consisted of 4 polymide-coated nichrome twisted...” The
word “is” can be removed and I believe the word “wire” should be inserted between “nichrome” and
“twisted”.

We have corrected the text.

2. It seems that a more accurate title would be that thalamic and cortical projections differentially
“contribute to” striatal sound representations, rather than control.

We agree with this suggestion and have replaced the word to “contribute to” in the title.

Reviewer #3

*In the manuscript Liang Chen and colleagues use a combination of behaviour, pathway specific chemo-*
*genetics and in vivo electrophysiology to demonstrate that the thalamo-striatal projections from the dorsal*
*portion of the medial geniculate nucleus to the striatum is critical for performance on an auditory*
*discrimination task. They go on to show that the cortical and thalamic inputs to the auditory striatum, play*
*a different role in shaping the striatal sound responses. The writing is clear and their conclusions are*
*supported by their data. I just have one main concern that may affect the interpretation of the results,*
*otherwise I think the manuscript offers a very interesting contribution to the field and paves the way for*
*more investigation of the thalamo-striatal projection which has been neglected.*

We thank the reviewer for the helpful comments and organized the responses below.

*1. My main concern is to do with potential discrepancies in the amount of the auditory cortex and MGd*
*that will be infected with the injections of the AAV-ArchT. Presumably a large proportion of the MGd is*
*covered in their ArchT injections, whether the whole auditory cortex is covered is not clear. This could*
*have consequences for the interpretation of their results as if a smaller proportion of the auditory cortex is*
*infected this could lead to an underestimate of the cortical contribution to auditory responses in the*
*striatum. In addition, could not covering the whole of the auditory cortex also potential lead to*
*observations that the peak response is preferential suppressed? To start to address this the authors*
*could at least show the spread and cortical coverage of their ArchT injections. In addition, if the injections*
*did not cover the entire cortical area I would suggest that a few animals are added to the study where a*
*large cortical injections covering all if not more than the auditory cortex are performed.*

We agree that the infection coverage is important for our experiments, analyses and
interpretation. In response to this concern: 1) we included three example images from
one animal in **Suppl. Fig. 7** to demonstrate the infection coverage; 2) we emphasized
that in our analysis we only included animals that were confirmed for adequate ArchT
expression in either the primary auditory cortex or MGB; and 3) as commented by this
reviewer, and also given the nature of viral labeling (varied cells labeled from animal to
animal), we added the following statement into our discussion: “Both the primary
auditory cortical neurons and their projections to the auditory striatum are tonotopically
organized (Xiong, Znamenskiy et al. 2015). If inhibition of only part of the primary
auditory cortex leads to the preferential suppression of the peak response, we would
expect that the striatal neurons showing suppression effects would share the same best
frequencies. However, within one animal we identified tuning effects from MSNs with
very different best frequencies, which suggests that the preferential suppression of peak
response was not due to insufficient cortical infection coverage.” (Page 13 lines 3-8)

*2. I think it would also be beneficial to show how the different MSN response types (sustained, onset,*
*offset and suppression) are effected by the inhibition. It is not clear from the bulk analysis if the tuning of*
*each response class is equally affected by the optical inhibition. This is important as in the individual*
*examples different response types are shown.*

We have now included two supplementary figures to demonstrate the inhibition effects
on different MSN response types (**Suppl. Figs. 8&9**). The turning curves of sustained
and onset groups showed similar changes in response to inhibition of projections from
MGB or the primary auditory cortex (**Suppl. Fig. 8**). We only have 3 offset-responsive
MSNs, one from a control mouse, one from MGB inhibition mouse, and one from
cortical inhibition mouse (**Suppl. Fig. 9A**). Thus, we cannot draw a conclusion on how
inhibition affects their tuning properties. The projection inhibition has no effect on
suppression type of MSN (**Supple. Fig. 9B**), and we have included these results in our
discussion (Page 13 lines 21-23 & page 14 lines 1-3).

*Minor points include,*
*3. Figure 1. Could the authors check if the images have been reversed as the red image looks like a*
*cortical injection as the axonal tracks characteristic of cortical fibres passing through the striatum are*
*present. This may also be a feature of MGd-striatal projections but I wanted to check.*

We thank the reviewer for pointing out this interesting feature. We have confirmed that
the color labels are correct in these images.

*4. Figure 3,4,5. In panel A for all figures AchT should be replaced with ArchT.*

We have corrected it.

*5. Page 4 paragraph 3, it would be helpful to include how the electrical stimulation is performed in the*
*main text. i.e local electrical stimulation within the striatum.*

We have included the detailed methods for local stimulations in the main text. In the
revised manuscript, we used ChR2-mediated terminal stimulation to validate CNO-
hM4Di inhibition (Page 4 lines 20-23).

*6. Page 7 paragraph 3, the authors mention that the offset and suppression MSN response types are*
*generally not frequency tuned, yet the two example neurons appear to have a frequency preference.*
*Could this be quantified for all responses types in a supplemental figure?*

According to our tuning analysis and definition (**Methods**, page 20 lines 15-20), we
agree with the reviewer that it was not accurate to say offset MSNs were not frequency
tuned based on only 3 neurons, whereas the suppression MSNs were not frequency
tuned (**Suppl. Fig. 9**). We have corrected the text and included quantifications of tone-
response and tuned populations (Page 7 lines 9-15 & page 8 lines 5-8).

References

- Berke, J. D. (2008). "Uncoordinated firing rate changes of striatal fast-spiking interneurons during
behavioral task performance." *Journal of Neuroscience* **28**(40): 10075-10080.
- Gage, G. J., C. R. Stoetznner, A. B. Wiltschko and J. D. Berke (2010). "Selective activation of striatal fast-
spiking interneurons during choice execution." *Neuron* **67**(3): 466-479.
- Gomez, J. L., J. Bonaventura, W. Lesniak, W. B. Mathews, P. Sysa-Shah, L. A. Rodriguez, R. J. Ellis, C. T.
Richie, B. K. Harvey, R. F. Dannals, M. G. Pomper, A. Bonci and M. Michaelides (2017). "Chemogenetics
revealed: DREADD occupancy and activation via converted clozapine." *Science* **357**(6350): 503-507.
- Graveland, G. A. and M. DiFiglia (1985). "The frequency and distribution of medium-sized neurons with
indented nuclei in the primate and rodent neostriatum." *Brain Res* **327**(1-2): 307-311.
- Moore, A. K. and M. Wehr (2013). "Parvalbumin-Expressing Inhibitory Interneurons in Auditory Cortex
Are Well-Tuned for Frequency." *Journal of Neuroscience* **33**(34): 13713-+.
- Spellman, T., M. Rigotti, S. E. Ahmari, S. Fusi, J. A. Gogos and J. A. Gordon (2015). "Hippocampal-
prefrontal input supports spatial encoding in working memory." *Nature* **522**(7556): 309-+.
- Stujenske, J. M., T. Spellman and J. A. Gordon (2015). "Modeling the Spatiotemporal Dynamics of Light
and Heat Propagation for In Vivo Optogenetics." *Cell Reports* **12**(3): 525-534.
- Xiong, Q., P. Znamenskiy and A. M. Zador (2015). "Selective corticostriatal plasticity during acquisition of
an auditory discrimination task." *Nature* **521**(7552): 348-351.
- Zemanick, M. C., P. L. Strick and R. D. Dix (1991). "Direction of transneuronal transport of herpes simplex
virus 1 in the primate motor system is strain-dependent." *Proc Natl Acad Sci U S A* **88**(18): 8048-8051.

Reviewers' Comments:

Reviewer #1:

Remarks to the Author:

Most of the points that I raised have been adequately addressed and the paper has been significantly improved. However, I still have a few concerns that ought to be addressed.

The authors' presentation of the retrograde tracing experiments, although improved, is still not adequate (suppl fig 5). The figure now contains a panel that is not cut off at the bottom of MGBv and reveals a significant number of thalamo-striatal neurons ventral of MGv, i.e. in what is labelled as SPFP and PP. In fact, it appears that here (in contrast to suppl fig 1) we have more green neurons SPFP/PP than in the MGD. However, these structures are not included in the analysis. What structure then are these neurons attributed to. Furthermore, how do the authors distinguish between the different higher-order nuclei of the auditory thalamus, i.e. MGD, MGm and SG? I'm not saying that it is particularly important to distinguish between them, especially as the borders are by no means obvious -in my opinion it would suffice here to distinguish between first order (MGBv) and higher order nuclei (i.e. all other nuclei) - but if the authors do make these distinctions it is important to state on what basis they do this.

I'm concerned with the analysis that indicates a reduction in firing rate at the best frequency from control to light condition (Fig 3D,E; 4D,E; 5D,E). Apologies for not spotting this in the initial review but recording a tuning curve, measuring the firing rate at its peak, i.e. at the best frequency, and comparing that firing rate with the firing rate at the same frequency of a second tuning curve, is a method that is inherently biased to finding a reduction in firing rate from the first to the second tuning curve, especially when dealing with noisy data (and the tuning curves of the striatal neurons do appear rather noisy). At worst this could give the impression of a firing rate reduction where there actually is none. A better way of conducting this analysis is to pool the data from both (control and light) conditions and take the BF and peak firing rate from this pooled tuning curve. The firing rate at BF during control and light conditions is then compared to the value from the pooled tuning curve. Alternatively, the authors could split their control data in half, yielding a control tuning curve A and B for each neuron, and perform their analysis on these two tuning curves to find out how much of reduction they would expect to see by chance.

page 8, line 21: "while those projecting" should, I assume, read "while those neurons projecting".

page 20/21, line 23/1: "into the MGB." Shouldn't this say "...MGB and primary auditory cortex"?

Reviewer #2:

Remarks to the Author:

Chen, Wang, Ge, and Xiong have addressed all of the major concerns that I presented in my original review. As I mentioned then, I believe this is a nicely executed and well-written set of experiments that help us clarify how auditory and motor-related systems in the brain interact with one another to guide behavior.

Reviewer #3:

Remarks to the Author:

The authors have addressed all my concerns and I now find the manuscript suitable for publication.

We thank again the reviewers for their careful reading, and R#1 for additional constructive remarks. We have taken the suggestions and revised the manuscripts accordingly. Please find below a detailed point-by-point response to all comments (reviewer's comments in grey and italic, our reply in black).

Reviewer #1 (Remarks to the Author):

1. Most of the points that I raised have been adequately addressed and the paper has been significantly improved. However, I still have a few concerns that ought to be addressed. The authors' presentation of the retrograde tracing experiments, although improved, is still not adequate (suppl fig 5). The figure now contains a panel that is not cut off at the bottom of MGBv and reveals a significant number of thalamo-striatal neurons ventral of MGv, i.e. in what is labelled as SPFP and PP. In fact, it appears that here (in contrast to suppl fig 1) we have more green neurons SPFP/PP than in the MGd. However, these structures are not included in the analysis. What structure then are these neurons attributed to. Furthermore, how do the authors distinguish between the different higher-order nuclei of the auditory thalamus, i.e. MGd, MGm and SG? I'm not saying that it is particularly important to distinguish between them, especially as the borders are by no means obvious -in my opinion it would suffice here to distinguish between first order (MGBv) and higher order nuclei (i.e. all other nuclei) - but if the authors do make these distinctions it is important to state on what basis they do this.

We agree with the reviewer of the complexity of thalamic region. In the current study, we have been focusing on two main input structures (MGB and the primary auditory cortex), therefore we didn't include the SPFP and PP into our analyses (i.e. those labeled neurons in SPFP and PP are not included in structures listed in **suppl Fig 5 C&D**). It would be interesting to analyze them in future studies.

In response to addition of a statement on the basis of this anatomical analysis, we have included this statement in our **Methods** section (page 16 lines 5-6). We defined the borders between MGd, MGv, MGm and SG based on mouse brain atlas registration and area proportion estimation.

2. I'm concerned with the analysis that indicates a reduction in firing rate at the best frequency from control to light condition (Fig 3D,E; 4D,E; 5D,E). Apologies for not spotting this in the initial review but recording a tuning curve, measuring the firing rate at its peak, i.e. at the best frequency, and comparing that firing rate with the firing rate at the same frequency of a second tuning curve, is a method that is inherently biased to finding a reduction in firing rate from the first to the second tuning curve, especially when dealing with noisy data (and the tuning curves of the striatal neurons do appear rather noisy). At worst this could give the impression of a firing rate reduction where there actually is none. A better way of conducting this analysis is to pool the data from both (control and light) conditions and take the BF and peak firing rate from this pooled tuning curve. The firing rate at BF during control and light conditions is then compared to the value from the pooled tuning curve. Alternatively, the authors could split their control data in half, yielding a control tuning curve A and B for each neuron, and perform their analysis on these two tuning curves to find out how much of reduction they would expect to see by chance.

We thank the reviewer to point out this concern. As indicated in **Figure. 3D&G**, we did not observe a significant shift of the best frequencies between control and light conditions. If the reduction of firing rates at best frequencies was due to the shift of best frequencies in noisy recordings, we would expect to observe elevated firing rates at shoulder frequencies. However, the plots in **Figures. 3F & 4F** showed that there was no elevation, suggesting likely there was no significant shift of the best frequencies.

We agree that splitting the control in two halves can serve as a reference to how tuning curve analysis may be affected by the noise of the data. However, we think that pooling the data from both conditions may instead mask any effect on best frequencies, since the control and light trials are 1:1 ratio. We therefore used the same

data set for analysis in **Figure 3** and randomly separated the control trials for each testing frequency, and then plotted the tuning curves and normalized the firing rates to the peak firing rates from the first group. In the panel **A** of the figure below, we showed three representative tuning curves from three single units. Panel **B&C** showed that the averaged tuning curves and the best-frequency values are not significantly different between these two groups ($n=26$, $p>0.1$).

3. page 8, line 21: "while those projecting" should, I assume, read "while those neurons projecting". We have revised it as suggested.

4. page 20/21, line 23/1: "into the MGB." Shouldn't this say "...MGB and primary auditory cortex"? We have revised it as suggested.

Reviewer #2 (Remarks to the Author):

Chen, Wang, Ge, and Xiong have addressed all of the major concerns that I presented in my original review. As I mentioned then, I believe this is a nicely executed and well-written set of experiments that help us clarify how auditory and motor-related systems in the brain interact with one another to guide behavior. We appreciate the reviewer's comments.

Reviewer #3 (Remarks to the Author):

The authors have addressed all my concerns and I now find the manuscript suitable for publication.

We appreciate the reviewer's comments.

Reviewers' Comments:

Reviewer #1:

Remarks to the Author:

I appreciate the authors' efforts at trying to address my concerns regarding their analysis of the light-induced changes in firing rate. However, it appears that there is some misunderstanding as to the point I was trying to make. It is not about shifts in BFs rather than the noise present in any measurement of a tuning curve biasing one towards finding a pre-post reduction if the analysis is performed as in this manuscript. To better bring across my point and illustrate the inappropriateness of their analysis I attached some matlab code below. It simulates data sets with 'control' (pre) and 'light' (post) tuning curves with a moderate amount of noise and calculates the change in firing rate as done by the authors. If we run this simulation 1000 times we will get a significant ($p < 0.05$) reduction in firing rate roughly 800 times (i.e. 80% of the time) simply because of the bias inherent in this approach. If we run the analysis in the way I suggested, i.e. find the BF from the pooled pre and post tuning curves, we get a significant difference ($p < 0.05$) only about 5% of the time which is exactly what we would expect by chance.

Their panel C seems to, in fact, corroborate my concerns. The normalized firing rates of the second tuning curves (leaving out the one outlier) fall between about 0.1 and 1.3 of the first and while it is difficult to make out the individual data points the median most likely lies well below 1.

I doubt that the conclusions to be drawn from this manuscript would need to be changed if the authors carried out the appropriate analysis, as the effects would most likely still come out, if perhaps be slightly reduced in size. However, the manuscript would undoubtedly be improved, provide the readers with more confidence in the results and set better standards for others.

%%%%% Matlab Simulation

% set up a baseline tuning curve

```
TuningCurve=[0 1 1 2 2 3 2 2 1 1 0];
```

```
for j=1:1000; %run simulation 1000 times
```

% figure

```
% produce a data set of 26 'pre' and 'post' tuning curves to compare
```

```
for i=1:26;
```

```
% produce a pre tuning curve by adding some noise to the baseline
```

```
% tuning curve
```

```
NoiseFactor=2; %change this value to vary amount of noise in tuning curve
```

```
preTuningCurvePlusNoise=rand(1,11)*NoiseFactor+TuningCurve;
```

```
% find BF (i.e. peak)
```

```
[RateAtBF,BF]=max(preTuningCurvePlusNoise);
```

```
% note firing rate at BF
```

```
PreRateAtBF(i)=RateAtBF;
```

```
% produce a pre tuning curve by adding some noise to the baseline
```

```
% tuning curve
```

```
postTuningCurvePlusNoise=rand(1,11)*NoiseFactor+TuningCurve;
```

```

% note down the post firing rate at BF as a fraction of the pre rate at BF
PostOverPre(i)=postTuningCurvePlusNoise(BF)/RateAtBF;

% note post firing at BF
PostRateAtBF(i)=postTuningCurvePlusNoise(BF);

% subplot(5,6,i)
% hold all
% plot(preTuningCurvePlusNoise)
% plot(postTuningCurvePlusNoise)

[RateAtBFPooled,BFPooled]=max(mean([preTuningCurvePlusNoise;postTuningCurvePlusNoise]));

PostRateAtBFPooled(i)=postTuningCurvePlusNoise(BFPooled);

PreRateAtBFPooled(i)=preTuningCurvePlusNoise(BFPooled);

% note down the post firing rate at BF as a fraction of the pre rate at BF
PostOverPrePooled(i)=postTuningCurvePlusNoise(BFPooled)/preTuningCurvePlusNoise(BFPooled);

end;
% figure,hist(PostOverPre)

% [a,b,c]=ttest(PostOverPre-1)

%run a paired t-test on the post vs pre firing rate at BF
[a,b,c]=ttest(PostRateAtBF, PreRateAtBF);
% note down p-value
P(j)=b;

%run a paired t-test on the post vs pre firing rate at BFPooled
[d,e,f]=ttest(PostRateAtBFPooled, PreRateAtBFPooled);
% note down p-value
PPooled(j)=e;

% note down the difference between the post firing rate at BF and the pre rate at BF
PostMinusPre=PostRateAtBF-PreRateAtBF;

% note down the difference between the post firing rate at BF and the
% pre rate at BF (Pooled)
PostMinusPrePooled=PostRateAtBFPooled-PreRateAtBFPooled;

% calculate averages
AvgPostOverPre(j)=mean(PostOverPre);
AvgPostMinusPre(j)=mean(PostMinusPre);

AvgPostOverPrePooled(j)=mean(PostOverPrePooled);
AvgPostMinusPrePooled(j)=mean(PostMinusPrePooled);

end;

```

We thank again the reviewer #1 for the comment and suggestion in analysis of light-induced effects. Please see below our response to this concern (reviewer's comments in grey and italic, our reply in black).

Reviewer #1 (Remarks to the Author):

1. Reviewer #1 (Remarks to the Author):

I appreciate the authors' efforts at trying to address my concerns regarding their analysis of the light-induced changes in firing rate. However, it appears that there is some misunderstanding as to the point I was trying to make. It is not about shifts in BFs rather than the noise present in any measurement of a tuning curve biasing one towards finding a pre-post reduction if the analysis is performed as in this manuscript. To better bring across my point and illustrate the inappropriateness of their analysis I attached some matlab code below. It simulates data sets with 'control' (pre) and 'light' (post) tuning curves with a moderate amount of noise and calculates the change in firing rate as done by the authors. If we run this simulation 1000 times we will get a significant ($p < 0.05$) reduction in firing rate roughly 800 times (i.e. 80% of the time) simply because of the bias inherent in this approach. If we run the analysis in the way I suggested, i.e. find the BF from

the pooled pre and post tuning curves, we get a significant difference ($p < 0.05$) only about 5% of the time which is exactly what we would expect by chance.

Their panel C seems to, in fact, corroborate my concerns. The normalized firing rates of the second tuning curves (leaving out the one outlier) fall between about 0.1 and 1.3 of the first and while it is difficult to make out the individual data points the median most likely lies well below 1.

I doubt that the conclusions to be drawn from this manuscript would need to be changed if the authors carried out the appropriate analysis, as the effects would most likely still come out, if perhaps be slightly reduced in size. However, the manuscript would undoubtedly be improved, provide the readers with more confidence in the results and set better standards for others.

We appreciate this concern. In response to this concern, in our revised manuscript, we separately picked the best frequencies from control trials and light-on trials. We compared firing rates at their own best frequencies (i.e. for each recorded unit, we compared the max firing rate from control trials to the max firing rate from light-on trials). Under this analysis, our conclusions stay the same as previous ones. The **Figure 3D&E&H**, **Figure 4D-F**, corresponding figure legends (page 26 lines 10-12; page 27 lines 10-13) and method section (page 21 lines 1-4) have been revised accordingly.

We agree with the reviewer that from a statistical point of view it is bias to use BF drawn from control trials and compare firing rates at this BF between control and light-on trials. It would be ideal to have an objective way to pick the best frequency for each single unit under both conditions. However, given that the experiments had been designed to record striatal tone responses under two biological conditions (with or without MGB/cortical inputs), we feel that pooling control and light-on trials together will be reasonable only when we already know there is no significant difference in structures of the tuning curves under these two conditions. Therefore, we think this analysis (pooling control and light-on trials) may not deliver a clear message to our question: what is the difference between the tuning curves under the two conditions (with or without MGB/cortical inputs).

Reviewers' Comments:

Reviewer #1:

Remarks to the Author:

The new analyses shown in 3D,E and 4D,E address my concerns.

Is there a reason why the same analysis was not also applied to 5D,E?

I may have overlooked this information. Otherwise please provide the sample sizes for also mice, i.e. not just how many neurons went into the various analyses but also how many mice these neurons came from.

We thank again the reviewer #1 for the comment and suggestion in analysis of light-induced effects. Please see below our response to this concern (reviewer's comments in grey and italic, our reply in black).

Reviewer #1 (Remarks to the Author):

1. Is there a reason why the same analysis was not also applied to 5D,E?

We have updated the Figure 5D&E using the same analysis.

2. I may have overlooked this information. Otherwise please provide the sample sizes for also mice, i.e. not just how many neurons went into the various analyses but also how many mice these neurons came from.

We have updated the sample sized for mice.